# Absorb and Converge: Provable Convergence Guarantee for Absorbing Discrete Diffusion Models

**Yuchen Liang**[†][*], **Renxiang Huang**[‡][*], **Lifeng Lai**[‡], **Ness Shroff**[†], **Yingbin Liang**[†]

[†]The Ohio State University    [‡]University of California, Davis

## Abstract

Discrete state space diffusion models have shown significant advantages in applications involving discrete data, such as text and image generation. It has also been observed that their performance is highly sensitive to the choice of rate matrices, particularly between uniform and absorbing rate matrices. While empirical results suggest that absorbing rate matrices often yield better generation quality compared to uniform rate matrices, existing theoretical works have largely focused on the uniform rate matrices case. Notably, convergence guarantees and error analyses for absorbing diffusion models are still missing. In this work, we provide the first finite-time error bounds and convergence rate analysis for discrete diffusion models using absorbing rate matrices. We begin by deriving an upper bound on the KL divergence of the forward process, introducing a surrogate initialization distribution to address the challenge posed by the absorbing stationary distribution, which is a singleton and causes the KL divergence to be ill-defined. We then establish the first convergence guarantees for both the $\tau$-leaping and uniformization samplers under absorbing rate matrices, demonstrating improved rates over their counterparts using uniform rate matrices. Furthermore, under suitable assumptions, we provide convergence guarantees without early stopping. Our analysis introduces several new technical tools to address challenges unique to absorbing rate matrices. These include a Jensen-type argument for bounding forward process convergence, novel techniques for bounding absorbing score functions, and a non-divergent upper bound on the score near initialization that removes the need of early-stopping.

## 1 Introduction

The diffusion model is one of the key branches of generative models. Inspired by non-equilibrium statistical physics, it was first introduced in [1] and was subsequently refined and extended by [2]. In recent years, the diffusion model has achieved many breakthroughs in the generation tasks under both continuous state spaces [3, 4] and discrete state spaces [5, 6]. A growing body of work suggests that for discrete data such as natural language and graphs, *discrete* diffusion models offer greater advantages and more flexibility than their *continuous* counterparts [7, 8, 9].

Diffusion models typically include a forward noising diffusion process and a backward denoising process. Under the continuous-time formulation of *discrete* diffusion models, the forward process can be characterized by continuous-time Markov chains (CTMCs), with some specially designed rate matrix. Commonly used rate matrices include the uniform rate matrix, which leads to a uniform stationary distribution; and the absorbing rate matrix, which results in a singleton (absorbing) stationary distribution. The generation quality is typically highly sensitive to the choice of the CTMC rate matrix. It was first reported by [5] that using the absorbing rate yields better performance than using the uniform rate in terms of both perplexity and negative log-likelihood (NLL) for text generation tasks. This empirical advantage was further confirmed subsequently by [9, 10, 11], where consistent improvement was observed using the absorbing rate matrix. Moreover, [5] established

---

[*]These authors contributed equally to this work.

39th Conference on Neural Information Processing Systems (NeurIPS 2025).

| Sampling algorithm | Early-stopping? | Uniform Rate [15] | Absorbing Rate (this paper) |
|---|---|---|---|
| $\tau$-leaping | Yes | $\widetilde{\mathcal{O}}(d^2/\varepsilon)$ | $\widetilde{\mathcal{O}}(d/\varepsilon)$ |
| | No | $\widetilde{\mathcal{O}}(d^2/\varepsilon)$ | $\widetilde{\mathcal{O}}(d\gamma^{-1}/\varepsilon)$ |
| Uniformization | Yes | $\mathrm{Pois}\Big(\mathcal{O}\big(d\log(d/\varepsilon)$ $+d\log\delta^{-1}\big)\Big)$ | $\mathrm{Pois}\Big(\mathcal{O}\big(d\log\log(d/\varepsilon)$ $+d\log\delta^{-1}\big)\Big)$ |
| | No | N/A | $\mathrm{Pois}\Big(\mathcal{O}\big(d\log\log(d/\varepsilon)$ $+d\gamma^{-1}\big)\Big)$ |

Table 1: Comparison of convergence results in terms of number of steps. Here we list only comparable references with the uniform rate (under the same algorithm and sample space). Note that [15] assumes symmetric rate matrix, which does not include the absorbing rate matrix studied in this paper. Here $d$ is the data dimension, $\delta$ is the amount of perturbation due to early-stopping, $\varepsilon$ is the target accuracy in KL-divergence, and $\gamma$ describes the minimum relative likelihood of the mask state in the data distribution (see Assumption 4). Here $\mathrm{Pois}(\lambda)$ refers to a Poisson random variable with mean $\lambda$. The sample space for all the results here is $[S]^d$.

close relationships between the absorbing discrete diffusion models and other popular language modeling approaches, including BERT [12] and the conditional masked language model (CMLM) [13].

The superior performance of discrete diffusion models has sparked considerable theoretical interest in understanding their convergence properties. However, existing convergence guarantees have primarily focused on the *uniform* rate matrix, with various sampling approaches analyzed in this setting. These include the uniformization method [14, 15], sampling via piecewise solutions of the Kolmogorov equation at each discretized step [16, 17], and the $\tau$-leaping sampler [6, 15]. Among those studies, the uniform rate matrix is explicitly assumed in [14, 16, 17], and a symmetric rate matrix is considered in [15]. Notably, these studies do not address the setting involving an absorbing rate matrix.

In contrast, although discrete diffusion models with an *absorbing* rate matrix have demonstrated superior empirical performance [5, 9], there has been no theoretical analysis to characterize their convergence behavior to date. This gap in the literature motivates our present study.

## 1.1 Our Contributions

Our overall contribution in this paper is to provide the first theoretical convergence guarantee for discrete diffusion models under the *absorbing* rate matrix. This is further described in the following four parts:

1. **Convergence of the forward process:** To address the challenge of irregular KL-divergence under the absorbing stationary distribution (i.e., a singleton), we design a smooth surrogate distribution which is both close to this singleton and easy to sample from. We further show that the data distribution in the forward process converges exponentially fast to this surrogate distribution in terms of KL divergence. Different from previous approaches using log-Sobolev inequalities, we employ a Jensen-based technique which is applicable when the absorbing rate matrix is used. Our approach enables a well-controlled initialization error and prepares for further convergence analysis for the reverse process.

2. **Convergence guarantee under the absorbing rate matrix:** For the $\tau$-leaping sampler, we establish an upper bound on the KL divergence between the generated and target distributions, showing that $\varepsilon$ KL-divergence accuracy can be achieved with $\widetilde{\mathcal{O}}(d/\epsilon)$ steps. Notably, our convergence rate under the absorbing rate matrix is *linear* in the data dimension $d$, which improves upon the quadratic dependency in $d$ established for the uniform rate matrix with $\tau$-leaping in [15]. This result implies that, for the same number of sampling steps, the absorbing rate matrix yields smaller KL-divergence, which aligns well with empirical results found in [9, 10, 11]. Moreover, for the uniformization sampler, we show that $\varepsilon$ KL-divergence accuracy is achievable in expected $\mathcal{O}(d(\log\log(d/\varepsilon)+\log\delta^{-1}))$ steps.

This also improves the expected $\mathcal{O}(d(\log(d/\varepsilon) + \log \delta^{-1}))$ steps previously required under the uniform rate matrix, further showing advantages of absorbing discrete diffusion models.

3. **Convergence guarantee without early-stopping:** Furthermore, we provide an interesting case which removes the need for early-stopping for both the $\tau$-leaping and the uniformization samplers. Intuitively speaking, this can be satisfied when the $[\mathrm{MASK}]$ token is selected as one of the likely tokens in the given vocabulary. Compared to [14, 15], we show that early-stopping might not be necessary even when using the uniformization sampler.

4. **New techniques for bounding absorbing scores:** One key component in our study is to investigate the properties of the score function under the absorbing rate matrix. Upon obtaining the exact expression of the score, we provide upper and lower bounds both with and without early-stopping. We show that the absorbing score is more well-controlled than for the uniform case for a large diffusion time, which enables smaller expected steps using uniformization. We also show a non-diverging score upper bound for quite relaxed data distributions, which removes the need of early-stopping. These score properties might have independent interest for future studies on absorbing diffusion models.

## 1.2 Related Works on Absorbing Discrete Diffusion Models

The superiority of absorbing discrete diffusion models have been confirmed in many empirical experiments, including on text [5, 9, 10, 11, 18], image [5, 10], music [19], DNA sequence and chemical molecule [18, 20]. Meanwhile, there have been many empirical studies investigating an improved training objective particularly for absorbing discrete diffusion models. For example, [11] reparameterized the concrete score training objective to achieve efficient training and sampling, [10] investigated and improved the training objective as a weighted integral of cross-entropy loss, and [18] derived a Rao-Blackwellized objective to tighten the Evidence Lower-Bound (ELBO) and to reduce training variances. Note that all of these works, while impressive, include only empirical results. A theoretical understanding of the superiority of absorbing diffusion models is still lacking. We have provided a more detailed literature review in Appendix A.

## 2 Preliminaries of Discrete Diffusion Models

Discrete diffusion models consist of a forward and a reverse process over the *discrete* data space.

The **forward process** is commonly modeled as a continuous-time Markov chain (CTMC) over a discrete state space [6]. We consider the state space $[S]^d$, representing a $d$-dimensional token space where each token is drawn from a vocabulary of size $S$. Accordingly, the training data $x_0 \in [S]^d$ consists of $d$ tokens, with an associated probability mass function denoted by $q_0$. Let $Q_t \in \mathbb{R}^{S^d \times S^d}$ be the rate matrix governing the forward process, where $Q_t(x, y)$ specifies the rate of transition from state $x$ to state $y$, for all $x, y \in [S]^d$. Then, given the previous state $x$, the transition probability from $t - \Delta t$ to $t$ is given by:

$$q_{t|t-\Delta t}(y|x) = \mathbb{1}\{y = x\} + Q_t(x, y)\Delta t + o(\Delta t).$$

Here, $\mathbb{1}\{y = x\}$ is the indicator function which equals 1 if $y = x$ and 0 otherwise. Clearly, the non-diagonal entries $Q_t(x, y) \geq 0$ for $x \neq y$, and the diagonal entries $Q(x, x) \leq 0$. We further have that $Q_t(x, x) = -\sum_{y:y \neq x} Q_t(x, y)$. Equivalently, the marginal distribution $q_t$ satisfies the Kolmogorov forward equation as follows:

$$\frac{d}{dt}q_t(y) = \sum_{x \in [S]^d} Q_t(x, y)q_t(x) = Q_t^\mathsf{T} q_t.$$

Given a state $x \in [S]^d$, we denote $x^i \in [S]$ as the $i$-th token of $x$. To simplify computation, it is often assumed that each token propagates independently in the forward process [6, 9, 16]. This implies that the forward conditional distribution can be factorized as $q_{t|0}(x_t|x_0) = \prod_{i=1}^d q_{t|0}^i(x_t^i|x_0^i)$. We define the rate matrix for each token as $Q_t^{tok} \in \mathbb{R}^{S \times S}$. It is shown in [6] that under such a forward process,

$$Q_t(x, y) = \begin{cases} Q_t^{tok}(x^i, y^i) & \text{if only } x^i \neq y^i, \\ 0 & \text{otherwise.} \end{cases}$$

We assume that $Q_t$ is time-homogeneous, and thus $Q_t \equiv Q$ and $Q_t^{tok} \equiv Q^{tok}$.

In this work, we focus on the **absorbing rate matrix**, which results in a singleton state towards the end of the forward process. Specifically, we let $[\text{MASK}] \in [S]$ denote the mask state in the vocabulary. Write $m(x)$ $(\leq d)$ for the number of $[\text{MASK}]$ in vector $x$. We define the absorbing rate matrix as

$$Q^{tok} = \mathbf{1}_S e_{[\text{MASK}]}^{\mathsf{T}} - I_S,$$

where $\mathbf{1}_S$ is an all-1 vector of length $S$, and $e_i$ is a unit vector where only the $i$-th element is 1. In other words, there are only two cases where $Q^{tok}(a, b) \neq 0$. First, the diagonal elements $Q^{tok}(a, a) = -1$, $\forall a \in [S] : a \neq [\text{MASK}]$, which corresponds to the case where no change occurs when the token is not yet in the mask state. Second, for the column corresponding to $[\text{MASK}]$, $Q^{tok}(a, [\text{MASK}]) = 1$, $\forall a \in [S] : a \neq [\text{MASK}]$. This corresponds to the transition from a non-mask to the mask state.

The **reverse process** can be designed to be the exact time-reversal process of the above forward process with an initial distribution $\breve{q}_0 = q_T$ [6, 21]. In particular, [6] shows that the time-reversal process $\{\breve{q}_t\}$ is also a CTMC from $t = 0$ to $t = T$ with the reverse rate matrix given by

$$\breve{Q}_t(x, y) = Q_{T-t}(y, x) \frac{q_{T-t}(y)}{q_{T-t}(x)} \quad \forall x, y \in [S]^d \text{ such that } x \neq y. \tag{1}$$

Then, the marginal distribution satisfies that $\breve{q}_t = q_{T-t}$. Similarly, for the diagonal elements in the reverse matrix, $\breve{Q}_t(x, x) = -\sum_{y:y \neq x} \breve{Q}_t(x, y)$.

For continuous-space diffusion models, one generally defines the score function as $\nabla_x \log q_t(x)$. Unfortunately, this is not applicable for discrete-space diffusion models where the gradient is not defined. Alternatively, the discrete score function is defined as $s_t(y, x) = \frac{q_t(y)}{q_t(x)}$. In order to prevent the score function from blowing up around $t = 0$, one common approach is to employ early stopping in the time-reversal process by setting the terminal time to be $t = T - \delta$ with a small constant $\delta$. Otherwise, if early-stopping is not applied, we simply set $\delta = 0$. To estimate the score function $s_t(y, x)$, we can parameterize it via a neural network and learn an approximation $\hat{s}_t \approx s_t$. One popular training loss is the score entropy $\mathcal{L}_{SE}$ [9], which is given by

$$\mathcal{L}_{SE}(\hat{s}_t) = \mathbb{E}_{x_t \sim q_t} \sum_{y \neq x_t} Q_t(y, x_t) \left( \hat{s}_t(y, x_t) - s_t(y, x_t) - s_t(y, x_t) \log \frac{\hat{s}_t(y, x_t)}{s_t(y, x_t)} \right). \tag{2}$$

In practice, for tractable training, the denoising score entropy is usually used, which is a variant of the score entropy [9].

In this work, we analyze two **sampling methods** commonly studied in the literature: the $\tau$-leaping method [6, 15, 22] and the uniformization method [14, 16, 23]. To explain these two methods, since it is hard to directly sample from a continuous-time reversal process, we divide the total time horizon $[0, T - \delta]$ into $N$ small intervals, such that $t_0 = 0$ and $t_N = T - \delta$. Given the estimated score $\hat{s}_{T-t_k}$, we define the estimated reverse rate matrix as

$$\hat{Q}_{t_k}(x, y) = Q_{T-t_k}(y, x)\hat{s}_{T-t_k}(y, x).$$

In the $\tau$**-leaping** sampling method [6, 22], for a given $x_{t_k}$, the next state is given by $x_{t_{k+1}} = x_{t_k} + \sum_{i=1}^d \sum_{s=1}^S (s - x_{t_k}^i) P_{is} e_i$,[1] where $P_{is}$ is a Poisson random variable with mean $\hat{Q}_{t_k}(x_{t_k}, x_{t_k} + (s - x_{t_k}^i)e_i)(t_{k+1} - t_k)$. Intuitively, this method can approximate the sampling process by simultaneously applying all transitions in the time interval $[t_k, t_{k+1}]$. Equivalently, on each interval $[t_k, t_{k+1})$, $\tau$-leaping approximates the piecewise constant $\hat{Q}_t(x, y)$ with a proxy $\tilde{Q}_t(x, y)$ such that $\tilde{Q}_t(x_{t_k}, y) = \hat{Q}_t(x_{t_k}, y)$ [6].

The **uniformization** sampling method [23] has been proven to be able to exactly simulate the time-inhomogeneous CTMC by constructing a Poisson process with piecewise constant intensity $\{\lambda_k\}_{k=0,...,B-1}$ (thus comes the name uniformization). Here it is required that $\lambda_k \geq \sup_{x \in [S]^d, t \in [t_k, t_{k+1})} (-\hat{Q}_t(x, x))$. At each time interval $[t_k, t_{k+1})$, the number of transition times $M_k$ is first sampled from a Poisson random variable with mean $\lambda_k(t_{k+1} - t_k)$. Then, each transition

---

[1] In practice, an additional clipping step is necessary to avoid boundary crossing behaviors. As shown in [15], such a step does not affect the convergence rate given sufficiently small step-sizes (cf. [15, Remark A.13]).

time is drawn uniformly over $[t_k, t_{k+1})$ which forms a set $\{\sigma_i\}_{i=1,\ldots,M_k}$. Finally, for each of these transition times $\sigma_i$, each dimension of the current state $x$ is transitioned to $s\ (\neq x^i)$ with probability $\lambda_k^{-1}\hat{Q}_{\sigma_i}(x, x + (s - x^i)e_i)$.

## 3 Main Results

In this section, we provide the convergence results for discrete diffusion models with the absorbing rate matrix.

### 3.1 Initialization through Surrogate Distribution

The initialization error of the sampling process closely depends on the convergence rate of the forward process under the absorbing rate matrix. To this end, we first characterize the evolution of the conditional and marginal distributions in the forward process in the following lemma. The proof is deferred to Appendix C.

**Proposition 1.** *Fix any time $t > 0$ and dimension $i \in [d]$. Define the token transition probability matrix of $q_{t|0}^i$ as $P_{0,t}^i$ such that $P_{0,t}^i(a, b) = q_{t|0}^i(b|a)$ for $a, b \in [S]$. Then,*

$$P_{0,t}^i = (1 - e^{-t})\mathbf{1}_S e_{[\text{MASK}]}^\intercal + e^{-t}I_S.$$

*Accordingly, if we similarly define the overall transition probability matrix of $q_{t|0}$ as $P_{0,t}$, then*

$$P_{0,t} = \left[(1 - e^{-t})\mathbf{1}_S e_{[\text{MASK}]}^\intercal + e^{-t}I_S\right]^{\otimes d},$$

*where $\otimes$ represents the tensor product. Also, the marginal distribution $q_t$ satisfies*

$$q_t^\intercal = q_0^\intercal \left[(1 - e^{-t})\mathbf{1}_S e_{[\text{MASK}]}^\intercal + e^{-t}I_S\right]^{\otimes d}.$$

Intuitively, with the absorbing rate matrix, Proposition 1 shows that the probability for each non-mask token to still remain in its original state at time $t$ is $e^{-t}$. If the state of the token changes, the only possibility is to transition to $[\text{MASK}]$ (i.e., with probability $1 - e^{-t}$). Once the token enters the mask state, it stays there forever. Thus, with a sufficient large terminal time $T$, the marginal distribution $q_T$ converges to the stationary distribution, which is $(\boldsymbol{\delta}_{[\text{MASK}]})^{\otimes d}$.

One main challenge in the analysis under the *absorbing* rate is that if we select the stationary distribution $(\boldsymbol{\delta}_{[\text{MASK}]})^{\otimes d}$ for initialization, the initialization error will diverge in KL-divergence because of the $\log 0$ term introduced for any $x \in [S]^d$ such that $\exists i : x^i \neq e_{[\text{MASK}]}$ and that $q_T(x) > 0$. Such a problem does not exist for the previous studies of the *uniform-rate* case where the stationary distribution is the uniform distribution over the state space. To address such an issue, we design a surrogate initial distribution to avoid the singleton distribution:

$$p_{init} = \left[(1 - \epsilon_T)\boldsymbol{\delta}_{[\text{MASK}]} + \frac{\epsilon_T}{S - 1}\sum_{j \neq [\text{MASK}]}\boldsymbol{\delta}_j\right]^{\otimes d}, \tag{3}$$

where $\epsilon_T > 0$ is a small positive constant that vanishes as $T \to \infty$. Here, instead of the stationary distribution, the above surrogate initialization distribution is asymptotically a singleton that is located at $(\boldsymbol{\delta}_{[\text{MASK}]})^{\otimes d}$ as $T \to \infty$. For any finite $\epsilon_T$, a small mass is distributed equally across all non-mask states on each dimension. With such an initialization, the KL-divergence is bounded away from infinity as long as $\epsilon_T$ is finite. We then characterize its initialization error in the following theorem. The proof is given in Appendix D.

**Theorem 1.** *Consider the surrogate initialization distribution in Equation* (3)*, and let $\epsilon_T = e^{-T}$. Then we have*

$$\text{KL}(q_T || p_{init}) \lesssim de^{-T}.$$

**New analysis approach:** Our analysis is different from the existing approaches using log-Sobolev inequalities [14, 15, 16]. Specifically, it has been shown that if the rate matrix of the CTMC satisfies a modified log-Sobolev inequality [24, 25], then the initialization error (i.e., the mixing time) can be well controlled (i.e., having exponential decay). Verifying such modified log-Sobolev constant typically requires that the rate matrix is *symmetric*. This is not the case, however, for the absorbing

rate matrix, which is highly *asymmetric*. Instead, we use a Jensen-based approach similar to the case of continuous diffusion models in [26]. Specifically, the key is to decompose the KL divergence into the difference of the (negative) entropy of the forward conditional distribution and the (negative) cross-entropy between the conditional and the initialization distribution. Then, we immediately obtain an upper bound for the initialization error given the analytical form of the conditional distribution under the absorbing transition kernel (from Proposition 1). Notably, no extra assumption is required of the rate matrix. Our approach is not only more direct but also can be more generally applied to a wider class of rate matrices, including non-symmetric ones and those without known log-Sobolev constants. Meanwhile, our result might have independent interest for investigating the mixing properties of general CTMCs.

## 3.2 Convergence Guarantees with Early-Stopping

With the initialization distribution in (3), we are now ready to provide the convergence guarantees for both the $\tau$-leaping and uniformization methods. In this subsection, we focus on the setting with *early-stopping*, and will study that without *early-stopping* in Section 3.3.

For the $\tau$-leaping sampler, we adopt the following two assumptions, which have been commonly taken in the previous analyses under the uniform rate matrix [15, 16, 17].

**Assumption 1** (Score Estimation Error). The estimated score function $\hat{s}_{T-t_k}$ satisfies

$$\sum_{k=0}^{N-1}(t_{k+1}-t_k)\mathcal{L}_{SE}(\hat{s}_{T-t_k}) \leq \varepsilon_{score}.$$

**Assumption 2** (Bounded Score Estimate). There exists $M > 0$ such that $\forall x, y \in [S]^d$ with $Q_{T-t_k}(y,x) > 0$, the estimated score $\hat{s}_{t_k}$ satisfies $|\log \hat{s}_{T-t_k}(y,x)| \leq \log M, \forall k = 1, \ldots, N$.

Assumption 2 is commonly adopted in the previous studies for uniform-rate discrete diffusion models (e.g., [15, 16]). In practice, this can be satisfied with score-clipping during training [16]. Indeed, the convergence error bounds in our main results only at most depend on $\log M$.

The following theorem characterizes the convergence rate of $\tau$-leaping under absorbing rate matrix.

**Theorem 2.** *Suppose that $p_0 = p_{init}$ in (3) and $t_{k+1} - t_k = c \min\{1, T - t_k\}$. Also suppose that $m(x_0) \leq m_0 = \tilde{O}(1)$ almost surely. Then, under Assumptions 1 and 2, using the $\tau$-leaping sampler yields, we have, as $c, \delta \to 0$,*

$$\mathrm{KL}(q_\delta \| p_{T-\delta}) \lesssim de^{-T} + \varepsilon_{score} + d(T + \log(M\delta^{-1}))\frac{(T + \log \delta^{-1})^2}{N},$$

*where $\mathrm{TV}(q_0, q_\delta) \lesssim d\delta$. Thus, $\mathrm{KL}(q_\delta \| p_{T-\delta}) \leq \varepsilon$ if we choose $T = \log(d/\varepsilon)$ and $N = \tilde{\mathcal{O}}(d/\varepsilon)$.*

Theorem 2 provides the *first* convergence guarantee for *absorbing* discrete diffusion models using the $\tau$-leaping algorithm. Here, the target distribution $q_\delta$ is slightly perturbed from the true data distribution $q_0$ due to early-stopping.[2] Theorem 2 indicates that $\tilde{\mathcal{O}}(d/\varepsilon)$ steps are sufficient to reach this slightly-perturbed target distribution $q_\delta$ within an $\varepsilon$-error in KL-divergence. Compared with the state-of-the-art result of $\mathcal{O}(d^2)$ under the *uniform* rate in [15], our Theorem 2 shows an improved dependency in $d$ by a factor of $\mathcal{O}(d)$ under the *absorbing* rate matrix. Indeed, such an improvement is consistent with the empirical studies in [5, 9, 11], which shows an improved generation quality under the absorbing rate matrix compared to the uniform one. The complete proof is provided in Appendix E.

The key difference between our analysis and that under the uniform rate is on how to obtain upper and lower bounds for the score functions. As an example, if one naively applies the same technique in the uniform rate case, one would only obtain an upper bound for $s_t(y, x)$ that is exponential in $t$. Our key insight here is that instead of a uniform upper bound over all possible $x \neq y$ such that $x^j \neq y^j$ (cf. [16, Lemma 2] and [15, Assumption 4.4]), we only need an upper bound over those $x$ and $y$ such that $Q(y, x) > 0$, which, given the particular design of the absorbing rate matrix, is small for all $t > 0$. We have provided more details about the novelty of our approach in Section 4.

Next, we conduct the convergence analysis for the uniformization sampler. We adopt the following slightly modified estimation assumption, which is typically required in the previous analysis of the uniformization sampler [14, 15].

---

[2]Indeed, as shown in Lemma 2, the score function must blow up for certain cases when $t \to 0$. For these cases, a small perturbation around $t = 0$ is necessary.

**Assumption 3** (Uniform Score Estimation Error). The estimated score function $\hat{s}_{T-t}$ satisfies

$$\int_0^{T-\delta} \mathcal{L}_{SE}(\hat{s}_{T-t}) \mathrm{d}t \leq \varepsilon'_{score}.$$

**Theorem 3.** *Suppose that $\hat{s}_{T-t}(y,x) \asymp s_{T-t}(y,x)$ when $Q_{T-t}(y,x) > 0$, $t_{k+1} - t_k = c$ and $\lambda_k \lesssim \sup_{x \in [S]^d, t \in [t_k, t_{k+1})}(-\hat{Q}_t(x,x))$. Then, under Assumption 3, as $c, \delta \to 0$, we have*

$$\mathrm{KL}(q_\delta \| p_{T-\delta}) \lesssim de^{-T} + \varepsilon'_{score},$$

*where $\mathrm{TV}(q_0, q_\delta) \lesssim d\delta$. Thus, $\mathrm{KL}(q_\delta \| p_{T-\delta}) \leq \varepsilon$ by choosing $T = \log(d/\varepsilon)$, for which case $\mathbb{E}[N] = \mathcal{O}(d(\log\log(d/\varepsilon) + \log \delta^{-1}))$.*

Theorem 3 is the *first* convergence guarantee for *absorbing* discrete diffusion models under the uniformization sampler. For small enough $\delta$, in order to reach $\varepsilon$-level KL-divergence accuracy, Theorem 3 shows that the expected number of steps grows as $\mathcal{O}(\log\log \varepsilon^{-1})$. This improves that under the uniform rate, where $\mathcal{O}(\log \varepsilon^{-1})$ steps on average is required [14, 15]. The underlying reason lies in the score function $s_t$ when $t$ becomes large. For *uniform* CTMC, since the stationary distribution is uniform, $s_t$ is close to a constant for which a constant-level uniformization intensity is required. In comparison, for *absorbing* CTMC, since the stationary distribution is a singleton, $s_t$ decays as $t^{-1}$ for large $t$'s (see Lemma 1), which enables a much lower uniformization intensity and reduces the total expected number of steps. The proof of Theorem 3 is given in Appendix F.

### 3.3 Convergence Guarantees without Early-Stopping

While the early-stopping technique ensures theoretical guarantees, it comes at a cost of degraded sample quality. Indeed, even a small perturbation around $t = 0$ might introduce a large difference in the overall log-likelihood. In the following, we show that the early-stopping can be avoided for absorbing discrete diffusion models with the following assumption.

**Assumption 4.** Suppose that for all $i \in [d]$ and $x^{-i} \in [S]^{d-1}$,

$$\frac{q_0^i([\mathrm{MASK}] | x^{-i})}{\max_{a^i \in [S]: a^i \neq [\mathrm{MASK}]} q_0^i(a^i | x^{-i})} \geq \gamma > 0.$$

Here, Assumption 4 is made only on the initial data. This assumption can be justified as nearly necessary for the validity of the diffusion algorithm, as follows. By the second part of Lemma 2, if Assumption 4 is not satisfied, the score function will (nearly) diverge around $t = 0$. Since the algorithm relies on the score function to make progress at each step, such divergence at $t = 0$ would render the algorithm itself invalid in that regime. To satisfy Assumption 4, a sufficient condition is that $q_0$ has full support over $[S]^d$ (albeit without an explicit $\gamma$), i.e., when [MASK] corresponds to one of the existing tokens in the training data. Also note that Assumption 4 can be satisfied with a larger $\gamma$ when this chosen token becomes more likely.

**Comparison with other assumptions in the literature:** We compare Assumption 4 with two other assumptions in the existing literature under which the early stopping can be removed. Particularly, [16, Assumption 2] assumes that $q_0$ has full support and that the score $s_0(y,x)$ can be upper-bounded by a uniform constant for all $x$ and $y$ where only one component differs, and [15, Assumption 4.5] assumes some Lipschitz continuity condition for the score function when $t \approx 0$. Our Assumption 4 relaxes [16, Assumption 2] and only requires that $q_0$ to have full support. While Assumption 4 does not have a direct comparison with [15, Assumption 4.5], as justified above, it is (nearly) necessary to ensure the validity of the diffusion algorithm.

In the following, we provide the convergence guarantee for $\tau$-leaping sampler without early stopping.
**Theorem 4.** *Take $\delta = 0$. Suppose that Assumptions 1, 2 and 4 hold. Also suppose that $m(x_0) \leq m_0 = \tilde{\Theta}(1)$ almost surely. Then, choosing $t_{k+1} - t_k = c$, we have*

$$\mathrm{KL}(q_0 \| p_T) \lesssim de^{-T} + \varepsilon_{score} + \gamma^{-1}d(T + \log(M\gamma^{-1}))\frac{T^2}{N}.$$

*Thus, $\mathrm{KL}(q_0 \| p_T) \leq \varepsilon$ by choosing $T = \log(d/\varepsilon)$ and $N = \tilde{\mathcal{O}}\left(d\gamma^{-1}/\varepsilon\right)$.*

Therefore, when Assumption 4 is satisfied, Theorem 4 shows that we can exactly recover the data distribution without early-stopping, by taking constant step-sizes for $\mathcal{O}(d/\varepsilon)$ steps. Also note that

the number of required steps decreases as $\gamma$ increases. Intuitively speaking, the generation becomes faster when the chosen [MASK] token already occurs likely in the original data.

**Novel analysis approach:** The proof is given in Appendix G. One key component in the proof is to provide a non-diverging upper-bound on the score when $t \approx 0$. To this end, we first invoke the exact expression of $s_t$ (see (21)). Then, our key insight is that given an initial mask state, it will stay there for any $t > 0$, which guarantees that the denominator of $s_t(y, x)$ (which corresponds to $q_t(x)$ with at least one mask state in $x$) does not vanish for small $t$ (Lemma 6). Indeed, to strengthen this, we also show an almost[3] converse result to this: Suppose that [MASK] does not occur at all in the initial data, the score function must blow up when $t \approx 0$ (see second part of Lemma 2).

For the uniformization sampler, we also establish the convergence guarantee without early-stopping, whose proof is give in Appendix H.

**Theorem 5.** *Take $\delta = 0$. Suppose that Assumptions 3 and 4 hold. Then, choosing $\hat{s}_{T-t}(y, x) \asymp s_{T-t}(y, x)$ when $Q_{T-t}(y, x) > 0$, $t_{k+1} - t_k = c$ and $\lambda_k \lesssim \sup_{x \in [S]^d, t \in [t_k, t_{k+1})} (-\hat{Q}_t(x, x))$, and letting $c \to 0$, we have*

$$\mathrm{KL}(q_0 || p_T) \lesssim d e^{-T} + \varepsilon'_{score}.$$

*Thus, $\mathrm{KL}(q_0 || p_T) \leq \varepsilon$ by choosing $T = \log(d/\varepsilon)$ and $\mathbb{E}[N] = \mathcal{O}\left(d(\log\log(d/\varepsilon) + \gamma^{-1})\right)$.*

Theorem 5 is the *first* non-early-stopping result for the uniformization sampler. Note that for uniform CTMC, early-stopping is typically required to use the uniformization algorithm [14, 15]. The proof of Theorem 5 is straightforward by combining elements from Theorems 3 and 4.

## 4 Overview of Key Proof Techniques

In this section, we highlight the major novel elements in our proofs. We first focus on the case with early-stopping, and we identify any differences towards the end of this section. Given $\delta > 0$, the TV distance under absorbing rate matrix has an upper bound similar to the uniform-rate case (see Lemma 4). Thus, as follows, we focus on deriving the KL error bound for both the uniformization and the $\tau$-leaping samplers.

Following [15, Corollary 3.4], the error in the KL-divergence can be decomposed as

$$\mathrm{KL}(\breve{q}_{T-\delta} || p_{T-\delta}) \leq \mathrm{KL}(\breve{q}_0 || p_0) + \int_0^{T-\delta} \mathcal{L}_{SE}(\hat{s}_{T-t}) \mathrm{d}t.$$

For the uniformization method, if we assume uniform score estimation error as in Assumption 3, and since we can sample exactly from the time-inhomogeneous process induced by $\hat{s}_t$ using the uniformization method, then $\mathrm{KL}(q_\delta || p_{T-\delta}) \leq \mathrm{KL}(q_T || p_0) + \varepsilon'_{score}$. Meanwhile, the total number of steps is a Poisson r.v., whose mean satisfies that

$$\mathbb{E}[N] \lesssim \sum_{k=0}^{B-1} \sup_{x \in [S]^d, t \in [t_k, t_{k+1})} \left( \sum_{y: y \neq x} \hat{s}_{T-t}(y, x) Q(y, x) \right) (t_{k+1} - t_k). \tag{4}$$

For the case with $\tau$-leaping, an error corresponding to time-discretization will be introduced. Under Assumption 1 and following straight-forward error decomposition, the total error has an upper bound given by (where we have combined Equations (14) to (16))

$$\mathrm{KL}(\breve{q}_{T-\delta} || p_{T-\delta}) \lesssim \mathrm{KL}(\breve{q}_0 || p_0) + \varepsilon_{score} + \sum_{k=0}^{N-1} \int_{t_k}^{t_{k+1}}$$

$$\mathbb{E}_{x_{t_k} \sim \breve{q}_{t_k}} \sum_{y \neq x_{t_k}} \left| \log \frac{s_{T-t_k}(y, x_{t_k})}{\hat{s}_{T-t_k}(y, x_{t_k})} \right| |s_{T-t}(y, x_{t_k}) - \hat{s}_{T-t_k}(y, x_{t_k})| Q(y, x_{t_k}) \mathrm{d}t. \tag{5}$$

Thus, for both the uniformization method and the $\tau$-leaping algorithm, the results in Theorem 2–Theorem 5 can be established as long as (i) we have an exponentially-decaying upper bound for the initialization error under the absorbing rate matrix, and (ii) the score functions $s_t(y, x)$ have nice upper and lower bounds for $t \in [\delta, T]$. Here a lower bound is necessary because of the log operator. As follows, we provide the details of all these missing pieces.

**Convergence of Forward Process (Theorem 1):** In establishing the exponentially-decaying initialization error bound, we cannot directly invoke the log-Sobolev inequalities for the mixing time

---

[3]This becomes an exact converse when we further assume homogeneity in the data across each dimension, i.e., when $q_0^i(a) \equiv q_0^1(a)$ for all $i \in [d]$ and $a \in [S]$.

of a Markov chain (as in [14, 15, 16]) because the absorbing rate matrix does not have a known log-Sobolev constant. Instead, we decompose the KL-divergence as (Equations (7), (8) and (11)):

$$\text{KL}(q_T \| p_{init}) \leq \sum_{i=1}^{d} \mathbb{E}_{x_0^i \sim q_0^i} \mathbb{E}_{x_T^i \sim q_{T|0}^i(\cdot|x_0^i)} \left[ \log q_{T|0}^i(x_T^i|x_0^i) - \log p_{init}^i(x_T^i) \right],$$

where the last line follows from Jensen's inequality since $f(u) = u \log u$ is convex and the fact that the forward process is conditionally independent across the dimensions. Also, since we have the analytical form of $q_{t|0}^i$ from Proposition 1 (cf. Equation (9)) and the initialization distribution from Equation (3), the result is straight-forward. In particular, the exponential decay in $T$ is due to the fact that $q_{t|0}^i([\text{MASK}]|x_0^i) = e^{-t}$ for all $x_0^i \neq [\text{MASK}]$. Note that our approach is also applicable to the case with uniform rate matrix or more generally to any CTMC with conditionally independent rate. This highlights the generality of our approach.

**General Score Upper Bound (Lemma 1):** The upper bound on $s_t(y, x)$ is essential to further providing an upper bound for the number of steps using uniformization (see Equation (4)) and that for the discretization error using the $\tau$-leaping sampler (see Equation (5)). To this end, if we simply follow the technique for uniform rate (cf. [16, Lemma 2]), we would get (cf. Equation (20))

$$s_t(y, x) = \mathbb{E}_{x_0 \sim q_{0|t}(\cdot|x)} \left[ \frac{q_{t|0}^j(y^j|x_0^j)}{q_{t|0}^j(x^j|x_0^j)} \right] \leq \frac{1-e^{-t}}{e^{-t}}, \text{ for all } x \text{ and } y \text{ s.t. only } x^j \neq y^j.$$

This is problematic because the bound is *exponential* in $T$ for $t \in [\delta, T]$. Our key insight in the analysis is that instead of a uniform upper bound over all possible $x \neq y$ such that $x^j \neq y^j$, from Equations (4) and (5), we only need an upper bound over those $x$ and $y$ such that $Q(y, x) > 0$. Given the absorbing rate matrix, this is equivalent to the case where $x^j = [\text{MASK}]$ while $y^j \neq [\text{MASK}]$. Now, given that $Q(y, x) > 0$, the upper bound for $s_t(y, x)$ can be significantly improved as

$$s_t(y, x) = \frac{e^{-t}}{1-e^{-t}} \cdot q_{0|t}^j(y^j|x) \leq t^{-1}, \text{ for all } x \text{ and } y \text{ s.t. } Q(y, x) > 0.$$

Note that this upper bound decays as $t^{-1}$ for large $t$, which is much faster than under the uniform rate matrix (where the score is asymptotically a constant). This enables us to design a much lower intensity for the uniformization algorithm for large $t$'s, thus significantly reducing the total expected number of steps. Also, for $\tau$-leaping, this score upper bound is also essential to control the rate of change in the score and thus the term $|s_{T-t}(y, x_{t_k}) - \hat{s}_{T-t_k}(y, x_{t_k})|$ in (5) (see Lemma 3).

**General Score Lower Bound (Lemma 2):** The upper bound by itself is not sufficient for the analysis using $\tau$-leaping because of the $\log$ operator in Equation (5). For this reason, we also need to provide a score lower bound when $Q(y, x) > 0$, especially for the region where $s_t$ is small. From the expression of $s_t$, one key element is $q_{0|t}^j(y^j|x)$, which by Bayes' rule is equal to

$$q_{0|t}(y^j|x) = \frac{\sum_{u^{-j} \in [S]^{d-1}} q_0(u^{-j}, y^j) \cdot q_{t|0}(x^{M \setminus j}, x^{UM}, x^j | u^{-j}, y^j)}{\sum_{a^j \in [S]} \sum_{u^{-j} \in [S]^{d-1}} q_0(u^{-j}, a^j) \cdot q_{t|0}(x^{M \setminus j}, x^{UM}, x^j | u^{-j}, a^j)}. \tag{6}$$

Here we explicitly decompose $x$ into three different parts: (i) $x^{UM}$, which is the unmasked components in $x$, (ii) $x^{M \setminus j}$, which is the masked components except at the $j$-th one, and (iii) $x^j$, which is equal to $[\text{MASK}]$ since $Q(y, x) > 0$. Here, for each fixed $x_0 = (u^{-j}, a^j)$, only the conditional probability at the $j$-th element would differ for different $a^j$, which indicates that the lower bound is independent of $d$. Also, intuitively, in terms of $t$, this lower bound should decay no faster than the worst rate of the conditional probability, which is $e^{-t}$. Interestingly, for the case where $x_0^i \neq [\text{MASK}]$ *a.s.* for all $i \in [d]$, our approach would result in an improved lower bound, which diverges as $t \to 0$ at a rate that matches that of the upper bound (i.e., $t^{-1}$). This not only highlights the tightness of our bounds but also contributes to the general understanding of the score function, which might potentially be useful during training.

**Non-diverging Score Upper Bound (Lemma 6):** Now we consider the case where early-stopping can be removed. For both analyses using the uniformization and the $\tau$-leaping algorithms, the goal is to provide a non-diverging upper bound on $s_t(y, x)$ when $Q(y, x) > 0$ (see Equations (4) and (5)). Now, from the exact expression of $q_{0|t}(y^j|x)$ in (6), suppose that Assumption 4 holds, then the $q_0(u^{-j}, [\text{MASK}])$ terms in the denominator would introduce a constant lower bound *independent* of $t$ when $t$ is small. This would result in an upper bound of $s_t(y, x)$, which also does *not* depend on $t$.

## 5 Conclusion

In this paper, we have provided the first convergence rate analysis for discrete diffusion models under the absorbing rate matrix. We have first introduced a surrogate initialization distribution to address the challenge due to the ill-defined KL divergence. We have then established the first convergence guarantees for both the $\tau$-leaping and uniformization samplers, demonstrating improved rates over their counterparts using uniform rate matrices. Furthermore, under suitable assumptions, we have provided convergence guarantees without early-stopping. One future direction is to provide guarantees for the conditional generation of discrete diffusion models, where the absorbing rates would depend on the particular form of conditioning.

## Acknowledgments and Disclosure of Funding

The work of Y. Liang, N. Shroff and Y. Liang was supported in part by the U.S. National Science Foundation under the grants: NSF AI Institute (AI-EDGE) 2112471, DMS-2134145, CNS-2312836, CNS-2223452, CNS-2225561, and was sponsored by the Army Research Laboratory under Cooperative Agreement Number W911NF-23-2-0225. The work of R. Huang and L. Lai was supported in part by the U.S. National Science Foundation under the grants: CCF-2232907 and ECCS-2448268. The views and conclusions contained in this document are those of the authors and should not be interpreted as representing the official policies, either expressed or implied, of the Army Research Laboratory or the U.S. Government. The U.S. Government is authorized to reproduce and distribute reprints for Government purposes notwithstanding any copyright notation herein.

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

# Appendix

## A  Related Works

**Discrete diffusion model**
There have been plenty of empirical works on discrete diffusion models. [1] first proposed the concepts of the diffusion model with a non-equilibrium statistical physics framework, laying the theoretical foundations of the diffusion model. In addition to the continuous space diffusion, they also discussed the modeling and denoising of a binomial discrete diffusion process. Later, [27] proposed the Multinomial Diffusion model defined on categorical variables through a uniform transition kernel, pioneering the structure of directly modeling the discrete data. [5] introduced the Discrete Denoising Diffusion Probabilistic Model (D3PM) with the structured transition matrices to generalize the Multinomial Diffusion. It is also [5] that first proposed the absorbing discrete diffusion models. [6] embedded the discrete diffusion model into the Continuous-Time Markov Chain framework, modeling the forward and reverse processes as CTMCs and naturally deriving the continuous-time ELBO. They also adopted the $\tau$-leaping algorithm instead of the exact simulation to sample the reverse process, which reduced the computational cost in the high-dimensional setting. To learn the discrete diffusion model, [5] and [6] directly approximated the reverse kernel. [28] and [29] proposed ratio matching and concrete score matching, respectively. Subsequently, [9] constructed the Score Entropy Discrete Diffusion models (SEDDs) by introducing the score entropy as a score matching counterpart to continuous diffusion, to extend the score matching to the discrete field.

Discrete diffusion models have demonstrated comparable or better performance than continuous diffusion models. D3PM [5] outperformed continuous DDPM on the CIFAR-10 dataset regarding log-likelihood. SEDD [9] achieved lower perplexity than existing diffusion models in language modeling. Moreover, extensive empirical studies demonstrated the advantages of the discrete diffusion model

in tasks such as genomic sequence and protein design [30, 31, 32], image [6, 10, 33], music [6, 33], NLP [10, 34, 35, 18], and finite symmetric groups [36].

**Absorbing discrete diffusion model**
Beyond general discrete diffusion models, there have been several empirical studies that are particularly focused on the absorbing discrete diffusion models. [10] simplified the variational training objective as a weighted integral of cross-entropy. They also proposed a state-dependent masking schedule, which allows rate adjustment dynamically with states for better generation quality. [11] reparameterized the concrete score as the product of a time-dependent scalar and a time-independent conditional distribution. Through this reparameterization, they built a Reparameterized Absorbing Discrete Diffusion (RADD) model without time $t$ to achieve efficient training and sampling. Similar to [10], [18] parameterized the reverse posterior based on the structures of the absorbing state and derived a tighter continuous-time ELBO through Rao-Blackwellization. They also proposed a semi-autoregressive decoding method that allows sampling sequences of arbitrary length. [37] proposed an informed corrector for the absorbing diffusion models, for which they showed better performance than using the regular (uninformed) predictor-corrector scheme for masked models. Building upon [10, 18], [38] investigated the problem of fine-tuning an absorbing diffusion model by casting it as a Bayesian posterior sampling problem. They introduced the Discrete Denoising Posterior Prediction (DDPP) objective for efficient training and sampling from fine-tuned models. More recently, [39] further validated the better scalability of absorbing diffusion models than traditional autogressive models in language understanding tasks.

**Convergence analyses on discrete diffusion model**
[14] applied the sampling algorithm based on uniformization in the state space $\{0, 1\}^d$. Under the assumptions of score-entropy error and bounded score, they achieved a nearly linear dependence of the expected number of iterations on the dimension $d$. [16] performed analysis of a discrete-time sampling scheme in the state space $[S]^d$ via the Girsanov theorem. The work of [14] and [16] are focused on the uniform discrete diffusion models. Under the assumption of the symmetric rate matrix, [15] introduced a stochastic integral framework and first provided the error bound of KL divergence for the $\tau$-leaping algorithm. Note that all of the works above are not applicable to the *absorbing* discrete diffusion model, which is the main focus in this paper.

# B  List of Notations

We write $\mathbb{1}\{x = y\}$ as a function of $x$ and $y$ which equals 1 only if $x = y$. For $i = 1, \ldots, d$, we write $e_i$ is a vector where only the $i$-th element is 1 and other elements are 0's, and we write $\boldsymbol{\delta}_i$ as the distribution of a singleton whose p.m.f. is $e_i$. For a positive integer $S$, $[S] := \{1, \ldots, S\}$. Write $\mathbf{1}_S$ as a vector of length $S$ that contains all 1's, and $I_S$ as an identity matrix of size $S \times S$. Write $m(x)$ to denote the number of $[\text{MASK}]$ states in the vector $x$.

# C  Proof of Proposition 1

Recall that we have $Q^{tok} = \mathbf{1}_S e_{[\text{MASK}]}^{\mathsf{T}} - I_S$. Without loss of generality assume $[\text{MASK}] = S$, i.e., the last token in the vocabulary. First, we perform the eigen-decomposition of $Q^{tok}$ as

$$
\begin{aligned}
Q^{tok} &= P\Lambda P^{-1} \\
&= \begin{bmatrix} 1 & 0 & \cdots & 0 & 1 \\ 0 & 1 & \cdots & 0 & 1 \\ \vdots & \vdots & \ddots & \vdots & \vdots \\ 0 & 0 & \cdots & 1 & 1 \\ 0 & 0 & \cdots & 0 & 1 \end{bmatrix} \text{diag}(-1, \cdots, -1, 0) \begin{bmatrix} 1 & 0 & \cdots & 0 & -1 \\ 0 & 1 & \cdots & 0 & -1 \\ \vdots & \vdots & \ddots & \vdots & \vdots \\ 0 & 0 & \cdots & 1 & -1 \\ 0 & 0 & \cdots & 0 & -1 \end{bmatrix}.
\end{aligned}
$$

Note that

$$
\exp\left[\Lambda t\right] = \begin{bmatrix} \exp\left[-t\right] & 0 & \cdots & 0 \\ 0 & \exp\left[-t\right] & \cdots & 0 \\ \vdots & \vdots & \ddots & \\ 0 & 0 & \cdots & 1 \end{bmatrix}.
$$

Thus, solving the Kolmogorov forward equation, the transition probability matrix of the $i$-th token $x^i$ can be expressed as

$$P_{0,t}^i = \exp\left[\int_0^t Q^{tok}\mathrm{d}s\right] = P\exp\left[\Lambda t\right]P^{-1}$$

$$= \begin{bmatrix} \exp\left[-t\right] & 0 & \cdots & 0 & 1-\exp\left[-t\right] \\ 0 & \exp\left[-t\right] & \cdots & 0 & 1-\exp\left[-t\right] \\ \vdots & \vdots & \ddots & \vdots & \vdots \\ 0 & 0 & \cdots & \exp\left[-t\right] & 1-\exp\left[-t\right] \\ 0 & 0 & \cdots & 0 & 1 \end{bmatrix}_{S\times S}$$

$$= (1-e^{-t})\mathbf{1}_S e_{[\text{MASK}]}^{\mathsf{T}} + e^{-t}I_S.$$

Since each token propagates independently in each dimension, then $q_{t|0}(x_t|x_0) = \prod_{i=1}^{d} q_{t|0}^i(x_t^i|x_0^i)$, and

$$P_{0,t} = \left[P_{0,t}^i\right]^{\otimes d}$$

$$= \left[(1-e^{-t})\mathbf{1}_S e_{[\text{MASK}]}^{\mathsf{T}} + e^{-t}I_S\right]^{\otimes d}.$$

Hence, the marginal distribution $q_t$ at time $t$ is

$$q_t^{\mathsf{T}} = q_0^{\mathsf{T}}\left[(1-e^{-t})\mathbf{1}_S e_{[\text{MASK}]}^{\mathsf{T}} + e^{-t}I_S\right]^{\otimes d}.$$

## D   Proof of Theorem 1

The proof idea is adapted from that for the continuous diffusion model first in [26]. To start, we have

$$\text{KL}(q_T||p_{init}) = \sum_{x_T\in[S]^d} q_T(x_T)\log\frac{q_T(x_T)}{p_{init}(x_T)}$$

$$= \underbrace{\sum_{x_T\in[S]^d} q_T(x_T)\log q_T(x_T)}_{\text{Term}_1} - \underbrace{\sum_{x_T\in[S]^d} q_T(x_T)\log p_{init}(x_T)}_{\text{Term}_2}. \qquad (7)$$

We first focus on the first term in (7). Since

$$q_T(x_T) = \mathbb{E}_{x_0\sim q_0}\left[q_{T|0}(x_T|x_0)\right],$$

we have

$$\text{Term}_1 = \sum_{x_T\in[S]^d} \mathbb{E}_{x_0\sim q_0}\left[q_{T|0}(x_T|x_0)\right]\log\mathbb{E}_{x_0\sim q_0}\left[q_{T|0}(x_T|x_0)\right]$$

$$\overset{(i)}{\le} \sum_{x_T\in[S]^d} \mathbb{E}_{x_0\sim q_0}\left[q_{T|0}(x_T|x_0)\log q_{T|0}(x_T|x_0)\right]$$

$$= \mathbb{E}_{x_0\sim q_0}\left[\sum_{x_T\in[S]^d} q_{T|0}(x_T|x_0)\log q_{T|0}(x_T|x_0)\right]$$

$$\overset{(ii)}{=} \sum_{i=1}^{d} \mathbb{E}_{x_0^i\sim q_0^i}\left[\sum_{x_T^i\in[S]} q_{T|0}^i(x_T^i|x_0^i)\log q_{T|0}^i(x_T^i|x_0^i)\right] \qquad (8)$$

where $(i)$ follows by Jensen's inequality since $f(u) = u\log u$ is convex, and $(ii)$ follows because the negative entropy can be decomposed into a sum across each dimension when the transition kernel is independent. To proceed, we need to express the analytical solution for $q_{t|0}^i$. From Proposition 1,

$$q_{t|0}^i(z|a) = [(1-e^{-t})\mathbf{1}_S e_{[\text{MASK}]}^{\mathsf{T}} + e^{-t}I_S](a,z)$$

$$= \begin{cases} e^{-t} & \text{if } a = z \neq [\text{MASK}] \\ 1 - e^{-t} & \text{if } a \neq z = [\text{MASK}] \\ 1 & \text{if } a = z = [\text{MASK}] \\ 0 & \text{otherwise} \end{cases}. \tag{9}$$

Thus, using the convention that $0 \log 0 = 0$, we have

$$q_{T|0}^i(z|a) \log q_{T|0}^i(z|a) = \begin{cases} e^{-T} \log e^{-T} & \text{if } a = z \neq [\text{MASK}] \\ (1 - e^{-T}) \log (1 - e^{-T}) & \text{if } a \neq z = [\text{MASK}] \\ 0 & \text{otherwise} \end{cases}.$$

Then, when $x_0^i = a \neq [\text{MASK}]$, the negative entropy is

$$\sum_{x_T^i \in [S]} q_{T|0}^i(x_T^i|a) \log q_{T|0}^i(x_T^i|a)$$

$$= \sum_{z: z \neq [\text{MASK}]} q_{T|0}^i(z|a) \log q_{T|0}^i(z|a) + q_{T|0}^i([\text{MASK}]|a) \log q_{T|0}^i([\text{MASK}]|a)$$

$$\overset{(iii)}{=} q_{T|0}^i(a|a) \log q_{T|0}^i(a|a) + q_{T|0}^i([\text{MASK}]|a) \log q_{T|0}^i([\text{MASK}]|a)$$

$$= e^{-T} \log e^{-T} + (1 - e^{-T}) \log(1 - e^{-T}).$$

Here $(iii)$ follows because of the absorbing rate matrix. Otherwise, when $x_0^i = [\text{MASK}]$,

$$\sum_{x_T^i \in [S]} q_{T|0}^i(x_T^i|[\text{MASK}]) \log q_{T|0}^i(x_T^i|[\text{MASK}])$$

$$= q_{T|0}^i([\text{MASK}]|[\text{MASK}]) \log q_{T|0}^i([\text{MASK}]|[\text{MASK}])$$

$$= 0.$$

This yields that

$$\text{Term}_1 \leq \sum_{i=1}^d \mathbb{E}_{x_0^i \sim q_0^i} [\mathbb{1}\{x_0^i \neq [\text{MASK}]\}] \left( e^{-T} \log e^{-T} + (1 - e^{-T}) \log(1 - e^{-T}) \right)$$

$$\overset{(iv)}{=} \left( \sum_{i=1}^d (1 - q_0^i([\text{MASK}])) \right) \left( e^{-T}(-T) - e^{-T} + O(e^{-2T}) \right)$$

$$= \left( \sum_{i=1}^d (1 - q_0^i([\text{MASK}])) \right) e^{-T} (-T - 1) + O(de^{-2T}), \tag{10}$$

where $(iv)$ follows, by Taylor expansion, that $(1 - x) \log(1 - x) = -x + O(x^2)$ when $x$ is small.

Now, let us turn to the second term in (7). As follows we write $\epsilon = \epsilon_T$ for which we omit the $T$ dependency. Note that the specified $p_{init}$ has independent components, and

$$p_{init}^i(x^i) = \begin{cases} 1 - \epsilon, & \text{if } x^i = [\text{MASK}] \\ \frac{\epsilon}{S-1}, & \text{if } x^i \neq [\text{MASK}] \end{cases}.$$

Here, $\delta_i$ denotes a point mass distribution centered at state $i$. Thus,

$$\text{Term}_2 = \sum_{x_T \in [S]^d} q_T(x_T) \log p_{init}(x_T)$$

$$= \sum_{x_T \in [S]^d} \left[ \sum_{x_0 \in [S]^d} q_{T|0}(x_T|x_0) q_0(x_0) \right] \log p_{init}(x_T)$$

$$\overset{(v)}{=} \mathbb{E}_{x_0 \sim q_0} \mathbb{E}_{x_T \sim q_{T|0}(\cdot|x_0)} [\log p_{init}(x_T)]$$

$$\overset{(vi)}{=} \sum_{i=1}^d \mathbb{E}_{x_0 \sim q_0} \mathbb{E}_{x_T^i \sim q_{T|0}^i(\cdot|x_0)} [\log p_{init}^i(x_T^i)]$$

$$\overset{(vii)}{=} \sum_{i=1}^{d} \mathbb{E}_{x_0^i \sim q_0^i} \mathbb{E}_{x_T^i \sim q_{T|0}^i(\cdot|x_0^i)}[\log p_{init}^i(x_T^i)]$$

$$= \sum_{i=1}^{d} \mathbb{E}_{x_0^i \sim q_0^i} \left[ \mathbb{1}\{x_0^i \neq [\text{MASK}]\} \left( (1 - e^{-T}) \log(1 - \epsilon) + e^{-T} \log\left( \frac{\epsilon}{S-1} \right) \right) \right]$$

$$+ \sum_{i=1}^{d} \mathbb{E}_{x_0^i \sim q_0^i} \left[ \mathbb{1}\{x_0^i = [\text{MASK}]\} \right] \log(1 - \epsilon)$$

$$\overset{(viii)}{=} \sum_{i=1}^{d} \mathbb{E}_{x_0^i \sim q_0^i} \left[ \mathbb{1}\{x_0^i \neq [\text{MASK}]\} \left( (e^{-T} - 1)\epsilon + e^{-T} \log\left( \frac{\epsilon}{S-1} \right) \right) \right]$$

$$- \sum_{i=1}^{d} q_0^i([\text{MASK}])\epsilon + O(d\epsilon^2)$$

$$= \left( \sum_{i=1}^{d} (1 - q_0^i([\text{MASK}])) \right) \left[ (e^{-T} - 1)\epsilon + e^{-T} \log\left( \frac{\epsilon}{S-1} \right) \right]$$

$$+ \left( \sum_{i=1}^{d} (1 - q_0^i([\text{MASK}])) \right) \epsilon - d\epsilon + O(d\epsilon^2) \tag{11}$$

where $(v)$ follows by changing the order of summation, $(vi)$ follows because $p_{init}$ is independent across the dimensions, $(vii)$ follows because the forward process is conditionally independent, and $(viii)$ follows because by Taylor expansion, $\log(1 - x) = -x + O(x^2)$ when $x$ is small.

Now, combining (10) and (11) together, we have

$$\text{KL}(q_T || p_{init}) \lesssim d\epsilon + \left( \sum_{i=1}^{d} (1 - q_0^i([\text{MASK}])) \right) \cdot$$

$$\left( e^{-T} (-T - 1) - (e^{-T} - 1)\epsilon - e^{-T} \log\left( \frac{\epsilon}{S-1} \right) - \epsilon \right)$$

$$\leq d\epsilon + de^{-T} \left( (-T - 1) - \epsilon - \log\left( \frac{\epsilon}{S-1} \right) \right).$$

Now, we can choose $\epsilon = e^{-T}$. Thus,

$$\text{KL}(q_T || p_{init}) \lesssim de^{-T} + de^{-T} \left( (-T - 1) - e^{-T} - \log\left( \frac{e^{-T}}{S-1} \right) \right)$$

$$\lesssim de^{-T}.$$

## E   Proof of Theorem 2

First, using the Girsanov change-of-measure technique similar to [15, Corollary 3.4], we get

$$\text{KL}(\bar{q}_{T-\delta} || p_{T-\delta}) \leq \text{KL}(\bar{q}_{0:T-\delta} || p_{0:T-\delta})$$
$$= \text{KL}(\bar{q}_0 || p_0) +$$

$$\mathbb{E}_{x_{0:T-\delta} \sim \bar{q}_{0:T-\delta}} \left[ \int_0^{T-\delta} \sum_{y \neq x_t} \left( \hat{s}_{T-t}(y, x_t) - s_{T-t}(y, x_t) + s_{T-t}(y, x_t) \log \frac{s_{T-t}(y, x_t)}{\hat{s}_{T-t}(y, x_t)} \right) Q(y, x_t) \mathrm{d}t \right].$$

$$= \text{KL}(\bar{q}_0 || p_0) +$$

$$\sum_{k=0}^{N-1} \int_{t_k}^{t_{k+1}} \mathbb{E}_{x_t \sim \bar{q}_t} \left[ \sum_{y \neq x_t} \left( \hat{s}_{T-t_k}(y, x_t) - s_{T-t}(y, x_t) + s_{T-t}(y, x_t) \log \frac{s_{T-t}(y, x_t)}{\hat{s}_{T-t_k}(y, x_t)} \right) Q(y, x_t) \right] \mathrm{d}t$$

$$= \text{KL}(\bar{q}_0 || p_0) +$$

$$\sum_{k=0}^{N-1} \int_{t_k}^{t_{k+1}} \mathbb{E}_{x_t \sim \bar{q}_t} \left[ \sum_{y \neq x_t} \left( \hat{s}_{T-t_k}(y, x_t) + G(s_{T-t}(y, x_t); \hat{s}_{T-t_k}(y, x_t)) \right) Q(y, x_t) \right] dt$$

where we have defined that

$$G(x; y) := x \log \frac{x}{y} - x. \tag{12}$$

Note that $Q_{T-t} \equiv Q$ due to homogeneity. From Theorem 1, the initialization error term has an upper bound as

$$\mathcal{L}_{init} := \mathrm{KL}(\bar{q}_0 \| p_0) \lesssim de^{-T}.$$

Also note that the estimation error satisfies

$$\mathcal{L}_{est} := \sum_{k=0}^{N-1} (t_{k+1} - t_k) \cdot$$

$$\mathbb{E}_{x_{t_k} \sim \bar{q}_{t_k}} \sum_{y \neq x_{t_k}} \left( \hat{s}_{T-t_k}(y, x_{t_k}) + G(s_{T-t_k}(y, x_{t_k}); \hat{s}_{T-t_k}(y, x_{t_k})) \right) Q(y, x_{t_k}) \leq \varepsilon_{score}. \tag{13}$$

Thus, we can find the discretization error to be

$$\mathcal{L}_{disc} := \sum_{k=0}^{N-1} \int_{t_k}^{t_{k+1}} \mathbb{E}_{x_t \sim \bar{q}_t} \left[ \sum_{y \neq x_t} \left( \hat{s}_{T-t_k}(y, x_t) + G(s_{T-t}(y, x_t); \hat{s}_{T-t_k}(y, x_t)) \right) Q(y, x_t) \right] dt - \mathcal{L}_{est}$$

$$\overset{(i)}{=} \sum_{k=0}^{N-1} \int_{t_k}^{t_{k+1}} dt \cdot$$

$$\mathbb{E}_{\substack{x_t \sim \bar{q}_t \\ x_{t_k} \sim \bar{q}_{t_k}}} \sum_{y \neq x_t} G(s_{T-t}(y, x_t); \hat{s}_{T-t_k}(y, x_t)) Q(y, x_t) - G(s_{T-t_k}(y, x_{t_k}); \hat{s}_{T-t_k}(y, x_{t_k})) Q(y, x_{t_k})$$

$$= \sum_{k=0}^{N-1} \int_{t_k}^{t_{k+1}} dt \cdot$$

$$\left( \mathbb{E}_{\substack{x_t \sim \bar{q}_t \\ x_{t_k} \sim \bar{q}_{t_k}}} \sum_{y \neq x_t} G(s_{T-t}(y, x_t); \hat{s}_{T-t_k}(y, x_t)) Q(y, x_t) - \sum_{y \neq x_{t_k}} G(s_{T-t}(y, x_{t_k}); \hat{s}_{T-t_k}(y, x_{t_k})) Q(y, x_{t_k}) \right.$$

$$\left. + \mathbb{E}_{x_{t_k} \sim \bar{q}_{t_k}} \sum_{y \neq x_{t_k}} \left( G(s_{T-t}(y, x_{t_k}); \hat{s}_{T-t_k}(y, x_{t_k})) - G(s_{T-t_k}(y, x_{t_k}); \hat{s}_{T-t_k}(y, x_{t_k})) \right) Q(y, x_{t_k}) \right). \tag{14}$$

Here, for $(i)$, we note that $\hat{s}_{T-t_k}(y, x_t) Q(y, x_t) = \hat{s}_{T-t_k}(y, x_{t_k}) Q(y, x_{t_k})$ for the $\tau$-leaping algorithm.

We first focus on the first term in the discretization error, which is

$$\sum_{k=0}^{N-1} \int_{t_k}^{t_{k+1}} dt\, \mathbb{E}_{\substack{x_t \sim \bar{q}_t \\ x_{t_k} \sim \bar{q}_{t_k}}} \left[ \sum_{y \neq x_t} G(s_{T-t}(y, x_t); \hat{s}_{T-t_k}(y, x_t)) Q(y, x_t) \right.$$

$$\left. - \sum_{y \neq x_{t_k}} G(s_{T-t}(y, x_{t_k}); \hat{s}_{T-t_k}(y, x_{t_k})) Q(y, x_{t_k}) \right]$$

$$= \sum_{k=0}^{N-1} \int_{t_k}^{t_{k+1}} dt\, \mathbb{E}_{x_t \sim \bar{q}_t} \left[ \sum_{y \neq x_t} G(s_{T-t}(y, x_t); \hat{s}_{T-t_k}(y, x_t)) Q(y, x_t) \right.$$

$$\left. - \sum_{x_{t_k} \in [S]^d} q_{T-t_k | T-t}(x_{t_k} | x_t) \sum_{y \neq x_{t_k}} G(s_{T-t}(y, x_{t_k}); \hat{s}_{T-t_k}(y, x_{t_k})) Q(y, x_{t_k}) \right]$$

$$\overset{(i)}{\lesssim} -\sum_{k=0}^{N-1} \int_{t_k}^{t_{k+1}} dt\, (t - t_k) \mathbb{E}_{x_t \sim \bar{q}_t} \sum_{x_{t_k} \in [S]^d} Q(x_t, x_{t_k}) \sum_{y \neq x_t} G(s_{T-t}(y, x_t); \hat{s}_{T-t_k}(y, x_t)) Q(y, x_t)$$

$$\overset{(ii)}{\lesssim} \sum_{k=0}^{N-1} \int_{t_k}^{t_{k+1}} dt \, (t-t_k)d \cdot \mathbb{E}_{x_{t_k} \sim \bar{q}_{t_k}} \sum_{y \neq x_{t_k}} G(s_{T-t}(y, x_{t_k}); \hat{s}_{T-t_k}(y, x_{t_k}))Q(y, x_{t_k})$$

$$\overset{(iii)}{\lesssim} \sum_{k=0}^{N-1} (t_{k+1}-t_k)^2 d\mathcal{L}_{est} + d \sum_{k=0}^{N-1} \int_{t_k}^{t_{k+1}} dt(t-t_k) \cdot$$

$$\mathbb{E}_{x_{t_k} \sim \bar{q}_{t_k}} \sum_{y \neq x_{t_k}} \left( G(s_{T-t}(y, x_{t_k}); \hat{s}_{T-t_k}(y, x_{t_k})) - G(s_{T-t_k}(y, x_{t_k}); \hat{s}_{T-t_k}(y, x_{t_k})) \right) Q(y, x_{t_k})$$

$$\overset{(iv)}{=} o(\kappa \cdot (\varepsilon_{score} + \mathcal{L}_{disc})) \tag{15}$$

where $(i)$ follows because by definition of CTMC $q_{t|t-\Delta t}(y|x) \lesssim \delta_{y,x} + Q(x,y)\Delta t$, $(ii)$ follows because there are $O(d)$ non-zero terms in $Q(x_t, x_{t_k})$ and because $\bar{q}_t(x)/\bar{q}_{t_k}(x) = 1 + O(t-t_k)$ for each $x \in [S]^d$, $(iii)$ follows because of (13) and that $\hat{s}_{T-t_k}(y, x_{t_k})Q(y, x_{t_k}) > 0$ when $y \neq x_{t_k}$, and $(iv)$ follows as long as

$$\mathbb{E}_{x_{t_k} \sim \bar{q}_{t_k}} \sum_{y \neq x_{t_k}} \left( G(s_{T-t}(y, x_{t_k}); \hat{s}_{T-t_k}(y, x_{t_k})) - G(s_{T-t_k}(y, x_{t_k}); \hat{s}_{T-t_k}(y, x_{t_k})) \right) \cdot$$

$$Q(y, x_{t_k}) = O(t-t_k).$$

Hence, this term does not contribute to the overall upper bound in (14). Thus, it remains to upper-bound the second term in (14) for all $t \in [t_k, t_{k+1})$, in which the key is to upper-bound its integrand given by

$$\mathbb{E}_{x_{t_k} \sim \bar{q}_{t_k}} \sum_{y \neq x_{t_k}} \left( G(s_{T-t}(y, x_{t_k}); \hat{s}_{T-t_k}(y, x_{t_k})) - G(s_{T-t_k}(y, x_{t_k}); \hat{s}_{T-t_k}(y, x_{t_k})) \right) Q(y, x_{t_k})$$

$$\lesssim \mathbb{E}_{x_{t_k} \sim \bar{q}_{t_k}} \sum_{y \neq x_{t_k}} \left| \log \frac{s_{T-t_k}(y, x_{t_k})}{\hat{s}_{T-t_k}(y, x_{t_k})} \right| \cdot |s_{T-t}(y, x_{t_k}) - s_{T-t_k}(y, x_{t_k})| \, Q(y, x_{t_k}) \tag{16}$$

where the inequality comes from the fact that $G(x; y)$ is continuous and $\frac{\partial}{\partial x}G(x; y) = \log \frac{x}{y}$.

To proceed, we need to investigate $s_t$ under the absorbing rate. The following lemmas investigate some properties of $s_t$. Their proofs are in Appendix I.

**Lemma 1** (Score Upper Bound). *Fix $t > 0$ and $x \neq y$ such that $Q_t(y, x) > 0$. Let $j$ be the only index such that $x^j \neq y^j$. Then, $x^j = [\text{MASK}]$, and we have*

$$s_t(y, x) = \frac{e^{-t}}{1-e^{-t}} \cdot q_{0|t}^j(y^j|x) \leq t^{-1}.$$

**Lemma 2** (Score Lower Bound). *Fix $t > 0$ and $x, y \in [S]^d$. Given that $Q(y, x) > 0$ and that $q_t(y) > 0$, we have*
$$s_t(y, x) \gtrsim e^{-t}.$$
*Further, suppose that $q_0^i([\text{MASK}]) = 0$ for all $i \in [d]$, then a tighter lower bound can be applied:*

$$s_t(y, x) \gtrsim \frac{1}{e^t - 1}.$$

*Here note that $s_t(y, x)$ diverges at the same rate as does the upper bound as $t \to 0$.*

**Lemma 3** (Score Derivative Upper Bound). *Suppose that the number of masks in the data satisfies $m(x_0) \leq m_0 = \tilde{O}(1)$ almost surely. Given that $Q(y, x) > 0$, we have*

$$\left| \frac{\partial}{\partial t} s_t(y, x) \right| \lesssim \frac{e^t}{(e^t - 1)^2} \leq t^{-2}.$$

Let us now continue to upper-bound the second term in (14). With the score upper and lower bounds in Lemmas 1 and 2, a direct implication is that, when $Q(y, x) > 0$ and $q_t(y) > 0$,

$$|\log s_t(y, x)| \lesssim \max\{|\log t^{-1}|, |\log(e^{-t}) - \log S|\} \leq T + \log \delta^{-1} + \log S. \tag{17}$$

Without loss of generality assume that $q_{T-t_k}(y) > 0$.[4] Now, continuing (16), note that

$$G(s_{T-t}(y, x); \hat{s}_{T-t_k}(y, x)) - G(s_{T-t_k}(y, x); \hat{s}_{T-t_k}(y, x))$$

$$\lesssim \left| \log \frac{s_{T-t_k}(y, x)}{\hat{s}_{T-t_k}(y, x)} \right| \cdot |s_{T-t}(y, x) - s_{T-t_k}(y, x)|$$

$$\lesssim (T + \log \delta^{-1} + \log M) \, |s_{T-t}(y, x) - s_{T-t_k}(y, x)| \qquad (18)$$

where the last line follows from (17) and because $|\log \hat{s}_{T-t}(y, x)| \le \log M$ when $Q(y, x) > 0$ from Assumption 2.

Recall that $t \in [t_k, t_{k+1})$. Thus, an upper bound for the second term in (14) is

$$\mathbb{E}_{x_{t_k} \sim \bar{q}_{t_k}} \sum_{y \ne x_{t_k}} \left( G(s_{T-t}(y, x_{t_k}); \hat{s}_{T-t_k}(y, x_{t_k})) - G(s_{T-t_k}(y, x_{t_k}); \hat{s}_{T-t_k}(y, x_{t_k})) \right) Q(y, x_{t_k})$$

$$\overset{(i)}{\lesssim} (T + \log \delta^{-1} + \log M) \mathbb{E}_{x_{t_k} \sim \bar{q}_{t_k}} \sum_{y \ne x_{t_k}} |s_{T-t}(y, x_{t_k}) - s_{T-t_k}(y, x_{t_k})| \, Q(y, x_{t_k})$$

$$\lesssim (t - t_k)(T + \log \delta^{-1} + \log M) \mathbb{E}_{x_{t_k} \sim \bar{q}_{t_k}} \sum_{y \ne x_{t_k}} \sup_{t' \in (T-t_{k+1}, T-t_k]} \left| \frac{\partial}{\partial t'} s_{t'}(y, x_{t_k}) \right| Q(y, x_{t_k})$$

$$\overset{(ii)}{\lesssim} (t - t_k)(T + \log \delta^{-1} + \log M)(T - t_{k+1})^{-2} d$$

where $(i)$ follows from (18), and $(ii)$ follows from Lemma 3.

Combining all results for the discretization error, we arrive at

$$\mathcal{L}_{disc} \lesssim \sum_{k=0}^{N-1} \int_{t_k}^{t_{k+1}} \mathrm{d}t$$

$$\mathbb{E}_{x_{t_k} \sim \bar{q}_{t_k}} \sum_{y \ne x_{t_k}} \left( G(s_{T-t}(y, x_{t_k}); \hat{s}_{T-t_k}(y, x_{t_k})) - G(s_{T-t_k}(y, x_{t_k}); \hat{s}_{T-t_k}(y, x_{t_k})) \right) Q(y, x_{t_k})$$

$$\lesssim (T + \log \delta^{-1} + \log M) d \sum_{k=0}^{N-1} (t_{k+1} - t_k)^2 \max \left\{ 1, (T - t_{k+1})^{-2} \right\}.$$

Finally, to determine the parameter dependency in the summation, we can directly employ [26, Lemma 18] and get that when $t_{k+1} - t_k \le c \min \{1, T - t_k\}$, we have

$$\mathcal{L}_{disc} \lesssim (T + \log \delta^{-1} + \log M) d \frac{(T + \log \delta^{-1})^2}{N}.$$

Furthermore, taking $t_{k+1} - t_k = c \min \{1, T - t_k\}$, the number of steps satisfies that $N \lesssim c^{-1}(T + \log \delta^{-1})$. With this, we arrive that

$$\mathrm{KL}(\bar{q}_{T-\delta} || p_{T-\delta}) \lesssim d e^{-T} + \varepsilon_{score} + d(T + \log(M\delta^{-1})) \frac{(T + \log \delta^{-1})^2}{N}$$

as desired.

Finally, the following lemma, whose proof is in Appendix I, provides an upper bound in TV distance between $q_0$ and $q_\delta$. The proof is similar to the uniform-rate case as in [14, Theorem 6].

**Lemma 4.** *Under the absorbing rate function, we have*

$$\mathrm{TV}(q_0, q_\delta) \lesssim d\delta, \quad as \; \delta \to 0.$$

The proof of Theorem 2 is now complete.

---

[4]Indeed, if $q_s(y) > 0$, then by the absorbing rate property, we have $q_t(y) > 0$ for all $t > s$. This implies that if $q_{T-t_k}(y) = 0$, then $q_{T-t}(y) \equiv 0$ for all $t \in (t_k, t_{k+1}]$. For this case, trivially we have $G(s_{T-t}(y, x); \cdot) \equiv 0$ for all $t \in [t_k, t_{k+1}]$, where the difference is simply 0.

# F  Proof of Theorem 3

It is shown in [15] that uniformization can exactly simulate the reverse process without discretization error. Thus, from [15, Corollary 3.4], we have

$$\mathrm{KL}(\bar{q}_{T-\delta}||p_{T-\delta}) \leq \mathrm{KL}(\bar{q}_0||p_0)$$

$$+ \mathbb{E}_{x_{0:T-\delta} \sim \bar{q}_{0:T-\delta}} \left[ \int_0^{T-\delta} \sum_{y \neq x_t} \left( \hat{s}_{T-t}(y, x_t) - s_{T-t}(y, x_t) + s_{T-t}(y, x_t) \log \frac{s_{T-t}(y, x_t)}{\hat{s}_{T-t}(y, x_t)} \right) Q(y, x_t) \mathrm{d}t \right]$$

$$\lesssim d e^{-T} + \varepsilon'_{score},$$

where the last line follows from Theorem 1 and Assumption 3. Similarly to the proof of Theorem 2, note that we still have $\mathrm{TV}(q_0, q_\delta) \lesssim d\delta$ due to the early-stopping.

It now remains to determine the number of steps, which is usually a Poisson random variable due to simulating the CTMC process. Now, for each interval $[t_k, t_{k+1})$, uniformization requires that $\lambda_k \geq \sup_{x \in [S]^d, t \in [t_k, t_{k+1})} -\hat{Q}_t(x, x)$. As follows, we first provide an upper bound for $\lambda_k$ using the following lemma.

**Lemma 5.** *Fix $t > 0$ and $x$ such that $[\mathrm{MASK}] \in x$. Recall that $m(x)$ $(\leq d)$ is the number of $[\mathrm{MASK}]$ in $x$. Then,*

$$\sum_{y:y \neq x} s_t(y, x)Q(y, x) \leq m(x)\frac{e^{-t}}{1 - e^{-t}} \leq m(x)t^{-1}.$$

Thus, under the assumption that $\hat{s}_{T-t}(y, x) \asymp s_{T-t}(y, x)$ when $Q_{T-t}(y, x) > 0$, we have

$$-\hat{Q}_t(x, x) = \sum_{y:y \neq x} \hat{Q}_t(x, y) = \sum_{y:y \neq x} Q_{T-t}(y, x)\hat{s}_{T-t}(y, x) \lesssim d(T - t)^{-1}.$$

Note that different from the case under uniform rate, this upper bound is vanishing for large $(T - t)$'s.

Thus, since the sum of independent $\mathrm{Pois}(\lambda_k)$ r.v.s is distributed as $\mathrm{Pois}(\sum_k \lambda_k)$, the expectation of the total number of steps is given by

$$\mathbb{E}[N] = \sum_{k=0}^{B-1} \lambda_k(t_{k+1} - t_k)$$

$$\lesssim \sum_{k=0}^{B-1} d(T - t_{k+1})^{-1}(t_{k+1} - t_k).$$

Now, since we choose constant step-sizes as $t_{k+1} - t_k = c$,

$$\mathbb{E}[N] \lesssim d \sum_{k=0}^{B-1} \frac{1}{T - t_{k+1}}(t_{k+1} - t_k)$$

$$= d \sum_{k=0}^{B-1} \frac{1}{T - t_{k+1}} ((T - t_k) - (T - t_{k+1}))$$

$$\lesssim d \int_\delta^T \frac{1}{t} \mathrm{d}t$$

$$= d(\log T + \log \delta^{-1}).$$

Plugging in $T = \log(d/\varepsilon)$ completes the proof.

# G  Proof of Theorem 4

In order to provide convergence guarantees without-early stopping, we need to provide a tighter upper bound for the score function $s_t(y, x)$ (given $Q(y, x) > 0$), which does not diverge for small $t$'s. Thus, the following lemmas provide improved upper-bounds when Assumption 4 holds. The proof is in Appendix I.

**Lemma 6.** *Suppose that Assumption 4 holds. Fix $t \geq 0$ and $x \neq y$ such that $Q_t(y, x) > 0$. We have the following improved upper bound for $s_t$:*

$$s_t(y, x) \leq \min\left\{\frac{e^{-t}}{1 - e^{-t}}, \gamma^{-1}\right\} \leq \gamma^{-1}.$$

*In particular, this bound does not diverge as $t \to 0$.*

**Lemma 7.** *Suppose that Assumption 4 holds. Suppose that the number of masks in the data satisfies $m(x_0) \leq m_0 = \tilde{\Theta}(1)$ almost surely. Given that $Q(y, x) > 0$, we have*

$$\left|\frac{\partial}{\partial t} s_t(y, x)\right| \lesssim \min\left\{\frac{e^t}{(e^t - 1)^2}, \gamma^{-1}\right\} \leq \gamma^{-1}.$$

Also, note that the general lower bound in Lemma 2 still holds regardless of Assumption 4.

The rest of the proof is similar as Theorem 2, for which we provide an outline below. Now from Lemma 6, we have

$$|\log s_t(y, x)| \lesssim T + \log \gamma^{-1}, \quad \forall t \in [0, T].$$

Also, from Lemma 7, we have

$$|s_t(y, x) - s_s(y, x)| \lesssim \gamma^{-1}(t - s), \ \forall s < t.$$

Thus,

$$\mathbb{E}_{x_{t_k} \sim \bar{q}_{t_k}} \sum_{y \neq x_{t_k}} \left(G(s_{T-t}(y, x_{t_k}); \hat{s}_{T-t_k}(y, x_{t_k})) - G(s_{T-t_k}(y, x_{t_k}); \hat{s}_{T-t_k}(y, x_{t_k}))\right) Q(y, x_{t_k})$$

$$\lesssim (T + \log(M\gamma^{-1})) \mathbb{E}_{x_{t_k} \sim \bar{q}_{t_k}} \sum_{y \neq x_{t_k}} |s_{T-t}(y, x) - \hat{s}_{T-t_k}(y, x)| Q(y, x_{t_k})$$

$$\lesssim (t - t_k)(T + \log(M\gamma^{-1}))d\gamma^{-1}.$$

Continuing from (14), we further have

$$\mathcal{L}_{disc} \lesssim \sum_{k=0}^{N-1} \left(\int_{t_k}^{t_{k+1}} (t - t_k)dt\right) \cdot (T + \log(M\gamma^{-1}))d\gamma^{-1}$$

$$\asymp \gamma^{-1}d(T + \log(M\gamma^{-1})) \sum_{k=0}^{N-1} (t_{k+1} - t_k)^2.$$

Now, given $t_{k+1} - t_k \equiv c$ (or equivalently, $c = \frac{T}{N}$), we can derive that the total error is given by, without early-stopping,

$$\mathrm{KL}(q_0 || p_T) \lesssim de^{-T} + \varepsilon_{score} + \gamma^{-1}d(T + \log(M\gamma^{-1}))\frac{T^2}{N}.$$

## H   Proof of Theorem 5

The proof is straightforward by applying the modified score upper bound in Lemma 6 to the proof of Theorem 3. First, the upper bound for the total error is the same as in Theorem 3. From Lemma 6, when Assumption 4 holds, we have $s_t(y, x) \leq \gamma^{-1}$, and thus

$$-\hat{Q}_t(x, x) = \sum_{y:y \neq x} \hat{Q}_t(x, y) = \sum_{y:y \neq x} Q_{T-t}(y, x)\hat{s}_{T-t}(y, x) \lesssim d\min\{(T - t)^{-1}, \gamma^{-1}\}.$$

Therefore, the expectation of the number of steps satisfies that

$$\mathbb{E}[N] = \sum_{k=0}^{N-1} \lambda_{k+1}(t_{k+1} - t_k) \lesssim \sum_{k=0}^{N-1} d\min\{(T - t_{k+1})^{-1}, \gamma^{-1}\}(t_{k+1} - t_k)$$

$$\leq \sum_{k:T-t_k>1} d(T - t_{k+1})^{-1}(t_{k+1} - t_k) + \sum_{k:T-t_k\leq 1} d\gamma^{-1}(t_{k+1} - t_k)$$

$$\lesssim d(\log T + \gamma^{-1}).$$

Plugging in $T = \log(d/\varepsilon)$ yields the desired result.

# I  Proofs of Supporting Lemmas

## I.1  Proof of Lemma 1

Following a similar analysis as the proof of [16, Lemma 2] (and note that $t > 0$), we have

$$s_t(y, x) = \mathbb{E}_{x_0 \sim q_{0|t}(\cdot|x)} \left[ \frac{q_{t|0}^j(y^j|x_0^j)}{q_{t|0}^j(x^j|x_0^j)} \right]. \tag{19}$$

Let us now focus on this likelihood ratio. Recall the analytical expression for $q_{t|0}^i$ in (9). In light of the expectation operator in (19), as follows we only consider those $x_0$ and $x$'s such that $q_{0|t}(x_0|x) > 0$. Then,

$$
\frac{q_{t|0}^j(y^j|x_0^j)}{q_{t|0}^j(x^j|x_0^j)} \overset{(i)}{=}
\begin{cases}
\frac{q_{t|0}^j(y^j|x_0^j)}{e^{-t}} & \text{if } x_0^j = x^j \neq [\text{MASK}] \\
\frac{q_{t|0}^j(y^j|x_0^j)}{1-e^{-t}} & \text{if } x_0^j \neq x^j = [\text{MASK}] \\
q_{t|0}^j(y^j|x_0^j) & \text{if } x_0^j = x^j = [\text{MASK}]
\end{cases}
$$

$$
\overset{(ii)}{=}
\begin{cases}
\frac{1-e^{-t}}{e^{-t}} & \text{if } x_0^j = x^j \neq [\text{MASK}] \text{ and } y^j = [\text{MASK}] \\
\frac{e^{-t}}{1-e^{-t}} & \text{if } y^j = x_0^j \neq x^j = [\text{MASK}] \\
0 & \text{otherwise}
\end{cases}. \tag{20}
$$

Here in $(i)$ we only have three cases because if $x^j \neq [\text{MASK}]$ and $x_0^j \neq x^j$ (whether or not $x_0^j$ is [MASK] itself), we have $q_{t|0}(x|x_0) = 0 \implies q_{0|t}(x_0|x) = 0$. Also for $(ii)$ in order that $q_{t|0}^j(y^j|x_0^j) > 0$, we must have either $y^j = x_0^j$ or $y^j = [\text{MASK}]$. Meanwhile, we need to ensure that $y^j \neq x^j$.

Now, note that if we naively provide an upper bound (indeed, as with the uniform rate), we would get $s_t(y, x) \leq e^t - 1 \leq e^T - 1$. This is problematic and is due to the highly asymmetric design in the rate matrix.

Instead, let us now consider the condition that $Q(y, x) > 0$. By definition of the absorbing rate, since $Q_t(y, x) = Q^{tok}(y^j, x^j) > 0$, $x^j$ must be [MASK]. Thus, it is impossible to have $x^j \neq [\text{MASK}]$ yet $y^j = [\text{MASK}]$ (since by definition the state will stay at [MASK] after reaching there, making such $Q(y, x) = 0$). This implies that only the second non-zero case in (20) is applicable when $Q(y, x) > 0$. Plugging back into (19), we have, when $Q(y, x) > 0$,

$$s_t(y, x) = \frac{q_t(y)}{q_t(x)} = \frac{e^{-t}}{1-e^{-t}} q_{0|t}^j(y^j|x), \quad \text{such that } x^j = [\text{MASK}].$$

Therefore, we have

$$s_t(y, x) \leq \frac{e^{-t}}{1-e^{-t}} = \frac{1}{e^t - 1} \leq t^{-1}.$$

## I.2  Proof of Lemma 2

From Lemma 1, when $Q(y, x) > 0$, we have

$$s_t(y, x) = \frac{e^{-t}}{1-e^{-t}} \cdot q_{0|t}^j(y^j|x).$$

Here $j$ is the only index such that $y^j \neq x^j$.

As follows we explicitly express $q_{0|t}^j(y^j|x)$. To this end, we use the following notations. Given $x$, we write $x^M$ and $x^{UM}$ for the masked and unmasked tokens in $x$, respectively. Since $x^j = [\text{MASK}]$ (from Lemma 1), denote the masked tokens except $x^j$ as $x^{M\backslash j}$. For an arbitrary vector $u \in [S]^d$, write $u^{-j} \in [S]^{d-1}$ as its $j$-th element excluded. Also write $u^M$, $u^{UM}$, and $u^{M\backslash j}$ as the tokens in $u$ that corresponding respectively to $x^M$, $x^{UM}$, and $x^{M\backslash j}$. Also, denote the number of masked tokens in $x$ as $m(x)$, and $m(x) \in [1, d]$. We also slightly abuse the notation and write $q_{t|0}(y^i|x) = q_{t|0}^i(y^i|x)$.

Using Bayes' rule, we have

$$q_{0|t}(y^j|x) = \frac{q_{t,0}(x, y^j)}{q_t(x)} = \frac{q_{t,0}(x^M, x^{UM}, y^j)}{q_t(x^M, x^{UM})}$$

$$= \frac{\sum_{u^{-j} \in [S]^{d-1}} q_0(u^{-j}, y^j) \cdot q_{t|0}(x^M, x^{UM}|u^{-j}, y^j)}{\sum_{a^j \in [S]} \sum_{u^{-j} \in [S]^{d-1}} q_0(u^{-j}, a^j) \cdot q_{t|0}(x^M, x^{UM}|u^{-j}, a^j)}$$

$$= \left( \sum_{u^{M \backslash j} \in [S]^{m(x)-1}} q_0(u^{M \backslash j}, y^j, x^{UM}) e^{-t(d-m(x))} (1-e^{-t}) q_{t|0}(x^{M \backslash j}|u^{M \backslash j}) \right) \Big/$$

$$\left( \sum_{a^j : a^j \neq [\text{MASK}]} \sum_{u^{M \backslash j} \in [S]^{m(x)-1}} q_0(u^{M \backslash j}, a^j, x^{UM}) e^{-t(d-m(x))} (1-e^{-t}) q_{t|0}(x^{M \backslash j}|u^{M \backslash j}) + \right.$$

$$\left. \sum_{u^{M \backslash j} \in [S]^{m(x)-1}} q_0(u^{M \backslash j}, [\text{MASK}], x^{UM}) e^{-t(d-m(x))} q_{t|0}(x^{M \backslash j}|u^{M \backslash j}) \right).$$

Here the last line follows by the definition of the forward absorbing-rate process, which is conditionally independent and using the absorbing-rate probabilities. Note that $y^j \neq [\text{MASK}]$ and $x^j = [\text{MASK}]$. Thus, using Lemma 1, we have an analytical expression for the score:

$$s_t(y, x)$$

$$= \left( \sum_{u^{M \backslash j} \in [S]^{m(x)-1}} q_0(u^{M \backslash j}, y^j, x^{UM}) e^{-t(d-m(x))} \cdot e^{-t} \cdot q_{t|0}(x^{M \backslash j}|u^{M \backslash j}) \right) \Big/$$

$$\left( \sum_{a^j : a^j \neq [\text{MASK}]} \sum_{u^{M \backslash j} \in [S]^{m(x)-1}} q_0(u^{M \backslash j}, a^j, x^{UM}) e^{-t(d-m(x))} (1-e^{-t}) q_{t|0}(x^{M \backslash j}|u^{M \backslash j}) + \right.$$

$$\left. \sum_{u^{M \backslash j} \in [S]^{m(x)-1}} q_0(u^{M \backslash j}, [\text{MASK}], x^{UM}) e^{-t(d-m(x))} q_{t|0}(x^{M \backslash j}|u^{M \backslash j}) \right). \tag{21}$$

We first provide a general lower bound. Observe that in both the numerator and the denominator above, the time-dependent components and the $a^j$-varying components can be separated. Also, those time-dependent components are the same as long as $a^j \neq [\text{MASK}]$. Thus, continuing from (21) and noting that $1 - e^{-t} \leq 1$, we have

$$s_t(y, x)$$

$$\geq e^{-t} \left( \sum_{u^{M \backslash j} \in [S]^{m(x)-1}} q_0(u^{M \backslash j}, y^j, x^{UM}) e^{-t(d-m(x))} q_{t|0}(x^{M \backslash j}|u^{M \backslash j}) \right) \Big/$$

$$\left( \sum_{a^j \in [S]} \sum_{u^{M \backslash j} \in [S]^{m(x)-1}} q_0(u^{M \backslash j}, a^j, x^{UM}) e^{-t(d-m(x))} q_{t|0}(x^{M \backslash j}|u^{M \backslash j}) \right)$$

$$\geq e^{-t} \left( \sum_{u^{M \backslash j} \in [S]^{m(x)-1}} q_0(u^{M \backslash j}, y^j, x^{UM}) e^{-t(d-m(x))} q_{t|0}(x^{M \backslash j}|u^{M \backslash j}) \right) \Big/$$

$$\left( S \cdot \max_{a^j \in [S]} \sum_{u^{M \backslash j} \in [S]^{m(x)-1}} q_0(u^{M \backslash j}, a^j, x^{UM}) e^{-t(d-m(x))} q_{t|0}(x^{M \backslash j}|u^{M \backslash j}) \right)$$

$$\gtrsim S^{-1} e^{-t}.$$

The last line is explained as follows. Note that the numerator is strictly positive because $q_t(y) > 0$ and thus $s_t(y, x) > 0$. Then, for a set of non-negative numbers, any positive $c_k$ satisfies $\frac{c_k}{\max_k c_k} = \min\{1, \min_{c_{k'} : c_{k'} > c_k} \frac{c_k}{c_{k'}}\} > 0$. This yields the first result in the statement.

Next, we show an improved lower bound when $q_0^i([\text{MASK}]) = 0$ for all $i \in [d]$. Then, an implication is that

$$q_0(u^{M \backslash j}, [\text{MASK}], x^{UM}) = 0, \quad \forall j \in [d].$$

Thus, from (21), following a similar analysis,

$$s_t(y, x)$$

$$= \frac{e^{-t}}{1-e^{-t}} \left( \sum_{u^{M\backslash j} \in [S]^{m(x)-1}} q_0(u^{M\backslash j}, y^j, x^{UM}) e^{-t(d-m(x))} q_{t|0}(x^{M\backslash j}|u^{M\backslash j}) \right) \Big/$$

$$\left( \sum_{a^j : a^j \neq [\text{MASK}]} \sum_{u^{M\backslash j} \in [S]^{m(x)-1}} q_0(u^{M\backslash j}, a^j, x^{UM}) e^{-t(d-m(x))} q_{t|0}(x^{M\backslash j}|u^{M\backslash j}) \right)$$

$$= \frac{e^{-t}}{1-e^{-t}} \left( \sum_{\substack{u^{M\backslash j} \in [S]^{m(x)-1} \\ [\text{MASK}] \notin u^{M\backslash j}}} q_0(u^{M\backslash j}, y^j, x^{UM}) e^{-t(d-m(x))} q_{t|0}(x^{M\backslash j}|u^{M\backslash j}) \right) \Big/$$

$$\left( \sum_{a^j : a^j \neq [\text{MASK}]} \sum_{\substack{u^{M\backslash j} \in [S]^{m(x)-1} \\ [\text{MASK}] \notin u^{M\backslash j}}} q_0(u^{M\backslash j}, a^j, x^{UM}) e^{-t(d-m(x))} q_{t|0}(x^{M\backslash j}|u^{M\backslash j}) \right)$$

$$\gtrsim S^{-1} \frac{e^{-t}}{1-e^{-t}}$$

as claimed.

### I.3 Proof of Lemma 3

Throughout we employ the same set of notations as in Lemma 2. Given $x$, we write $x^M$ and $x^{UM}$ for the masked and unmasked tokens in $x$, respectively. Since $x^j = [\text{MASK}]$ (from Lemma 1), denote the masked tokens except $x^j$ as $x^{M\backslash j}$. For an arbitrary vector $u \in [S]^d$, write $u^{-j} \in [S]^{d-1}$ as its $j$-th element excluded. Also write $u^M$, $u^{UM}$, and $u^{M\backslash j}$ as the tokens in $u$ that corresponding respectively to $x^M$, $x^{UM}$, and $x^{M\backslash j}$. Also, denote the number of masked tokens in $x$ as $m(x)$, and $m(x) \in [1, d]$.

We consider the following two cases. First, suppose that $x_0^i \neq [\text{MASK}]$ almost surely for all $i \in [d]$. From [11, Theorem 1], we have

$$s_t(y, x) = \frac{e^{-t}}{1-e^{-t}} q_0^j(y^j|x^{UM}).$$

Here note that $q_0^j$ is a time-*independent* "clean-data" distribution. Thus, the time-derivative of the score function is equal to

$$\left| \frac{\partial}{\partial t} s_t(y, x) \right| = \left| \frac{\partial}{\partial t} \left( \frac{e^{-t}}{1-e^{-t}} \right) q_0^j(y^j|x^{UM}) \right| = \frac{e^t}{(e^t-1)^2} q_0^j(y^j|x^{UM}),$$

which implies that

$$\sup_{t' \in [s,t]} \left| \frac{\partial}{\partial t'} s_{t'}(y, x) \right| \leq \frac{e^s}{(e^s-1)^2} \leq s^{-2}.$$

Next, we consider the case where $[\text{MASK}]$ is possibly in the data. We first recall the analytical expression of $s_t(y, x)$ in (21):

$$s_t(y, x)$$

$$= \left( \sum_{u^{M\backslash j} \in [S]^{m(x)-1}} q_0(u^{M\backslash j}, y^j, x^{UM})(e^{-t})^{d-m(x)} \cdot e^{-t} \cdot (1-e^{-t})^{m(x)-1-m(u^{M\backslash j})} \right) \Big/$$

$$\left( \sum_{a^j : a^j \neq [\text{MASK}]} \sum_{u^{M\backslash j} \in [S]^{m(x)-1}} q_0(u^{M\backslash j}, a^j, x^{UM})(e^{-t})^{d-m(x)}(1-e^{-t})(1-e^{-t})^{m(x)-1-m(u^{M\backslash j})} + \right.$$

$$\left. \sum_{u^{M\backslash j} \in [S]^{m(x)-1}} q_0(u^{M\backslash j}, [\text{MASK}], x^{UM})(e^{-t})^{d-m(x)}(1-e^{-t})^{m(x)-1-m(u^{M\backslash j})} \right)$$

$$= \left( \sum_{u^{M\backslash j} \in [S]^{m(x)-1}} q_0(u^{M\backslash j}, y^j, x^{UM}) \cdot e^{-t} \cdot (1 - e^{-t})^{-m(u^{M\backslash j})} \right) \Bigg/$$

$$\left( \sum_{a^j : a^j \neq [\text{MASK}]} \sum_{u^{M\backslash j} \in [S]^{m(x)-1}} q_0(u^{M\backslash j}, a^j, x^{UM})(1 - e^{-t})(1 - e^{-t})^{-m(u^{M\backslash j})} + \right.$$

$$\left. \sum_{u^{M\backslash j} \in [S]^{m(x)-1}} q_0(u^{M\backslash j}, [\text{MASK}], x^{UM})(1 - e^{-t})^{-m(u^{M\backslash j})} \right).$$

By assumption, here $q_0(u^{M\backslash j}, [\text{MASK}], x^{UM}) > 0$ for some $u^{M\backslash j}$. Also, $m(u^{M\backslash j}) \leq m_0 = \tilde{O}(1)$. Observe that $s_t(y, x)$ is continuous in $t$. Taking the derivative of this ratio, we get $\frac{\partial}{\partial t} s_t(y, x) = T_1 - T_2$, where

$$T_1 := -\left( \sum_{u^{M\backslash j} \in [S]^{m(x)-1}} q_0(u^{M\backslash j}, y^j, x^{UM}) e^{-t} \cdot (1 - e^{-t})^{-m(u^{M\backslash j})} \frac{m(u^{M\backslash j}) + e^t - 1}{e^t - 1} \right) \Bigg/$$

$$\left( \sum_{a^j : a^j \neq [\text{MASK}]} \sum_{u^{M\backslash j} \in [S]^{m(x)-1}} q_0(u^{M\backslash j}, a^j, x^{UM})(1 - e^{-t})(1 - e^{-t})^{-m(u^{M\backslash j})} + \right.$$

$$\left. \sum_{u^{M\backslash j} \in [S]^{m(x)-1}} q_0(u^{M\backslash j}, [\text{MASK}], x^{UM})(1 - e^{-t})^{-m(u^{M\backslash j})} \right),$$

and

$$T_2 := -\left( \sum_{u^{M\backslash j} \in [S]^{m(x)-1}} q_0(u^{M\backslash j}, y^j, x^{UM}) \cdot e^{-t} \cdot (1 - e^{-t})^{-m(u^{M\backslash j})} \right) \cdot$$

$$\left( \sum_{a^j : a^j \neq [\text{MASK}]} \sum_{u^{M\backslash j} \in [S]^{m(x)-1}} q_0(u^{M\backslash j}, a^j, x^{UM})(m(u^{M\backslash j}) - 1)e^{-t}(1 - e^{-t})^{-m(u^{M\backslash j})} + \right.$$

$$\left. \sum_{u^{M\backslash j} \in [S]^{m(x)-1}} q_0(u^{M\backslash j}, [\text{MASK}], x^{UM}) \frac{m(u^{M\backslash j})e^{-t}}{1 - e^{-t}}(1 - e^{-t})^{-m(u^{M\backslash j})} \right) \Bigg/$$

$$\left( \sum_{a^j : a^j \neq [\text{MASK}]} \sum_{u^{M\backslash j} \in [S]^{m(x)-1}} q_0(u^{M\backslash j}, a^j, x^{UM})(1 - e^{-t})(1 - e^{-t})^{-m(u^{M\backslash j})} + \right.$$

$$\left. \sum_{u^{M\backslash j} \in [S]^{m(x)-1}} q_0(u^{M\backslash j}, [\text{MASK}], x^{UM})(1 - e^{-t})^{-m(u^{M\backslash j})} \right)^2.$$

Note the similarities in-between and with the expression of $s_t(y, x)$. Since $m(u^{M\backslash j}) \leq m_0$, we have

$$|T_1| \leq \frac{m_0 + e^t - 1}{e^t - 1} s_t(y, x)$$

$$|T_2| \leq \left( \frac{(m_0 - 1)e^{-t}}{1 - e^{-t}} + m_0 \frac{e^{-t}}{1 - e^{-t}} \right) s_t(y, x).$$

Since $m_0 = \tilde{O}(1)$ by assumption, using Lemma 1, we obtain that

$$\left| \frac{\partial}{\partial t} s_t(y, x) \right| \lesssim \frac{e^t}{(e^t - 1)^2} \leq t^{-2}.$$

Note that this bound is independent of $d$. The proof is now complete.

### I.4 Proof of Lemma 4

Write $\Pi(q_0, q_\delta)$ is the set of all joint probability measures with marginal distributions $q_0$ and $q_\delta$. Then,

$$\text{TV}(q_0, q_\delta) = \min_{\pi \in \Pi(q_0, q_\delta)} \mathbb{E}_{x_0, x_\delta \sim \pi} \mathbb{1}\{x_\delta \neq x_0\}$$

$$\leq \mathbb{E}_{x_0 \sim q_0} \left[ q_{\delta|0} \left( \{ x_\delta \neq x_0 \} | x_0 \right) \right]$$

$$\overset{(i)}{\leq} \sum_{i \in [d]} \mathbb{E}_{x_0^i \sim q_0^i} \left[ q_{\delta|0}^i \left( \{ x_\delta^i \neq x_0^i \} | x_0^i \right) \right]$$

$$\overset{(ii)}{=} \sum_{i \in [d]} \sum_{x_0^i \neq [\mathrm{MASK}]} q_0^i(x_0^i) q_{\delta|0}^i \left( [\mathrm{MASK}] | x_0^i \right)$$

$$\leq d \left( 1 - e^{-\delta} \right)$$

$$\asymp d\delta$$

where $(i)$ follows from the union bound, and $(ii)$ follows from the conditional probabilities under the absorbing rate.

### I.5 Proof of Lemma 5

From Lemma 1, when $Q(y, x) > 0$, we have

$$s_t(y, x) = \frac{e^{-t}}{1 - e^{-t}} \cdot q_{0|t}^j(y^j | x),$$

where $j$ is the only index such that $y^j \neq x^j = [\mathrm{MASK}]$. Thus,

$$\sum_{y:y \neq x} s_t(y, x) Q(y, x) = \sum_{y:Q(y,x)>0} s_t(y, x) Q(y, x)$$

$$\overset{(i)}{=} \sum_{j:x^j = [\mathrm{MASK}]} \sum_{y^j : y^j \neq [\mathrm{MASK}]} s_t(y, x)$$

$$\overset{(ii)}{=} \frac{e^{-t}}{1 - e^{-t}} \sum_{j:x^j = [\mathrm{MASK}]} \sum_{y^j : y^j \neq [\mathrm{MASK}]} q_{0|t}^j(y^j | x)$$

$$\leq \frac{e^{-t}}{1 - e^{-t}} m(x).$$

Here $(i)$ follows because the only positive entry in $Q(y, x)$ is 1, which corresponds to the case where $\mathrm{Ham}(y, x) = 1$, $y^j \neq [\mathrm{MASK}]$, and $x^j = [\mathrm{MASK}]$, and $(ii)$ follows by Lemma 1. The claim is thus established.

### I.6 Proof of Lemma 6

From Lemma 1, we already have an upper bound for $s_t(y, x)$ when $t$ is bounded away from 0, which is

$$s_t(y, x) \leq t^{-1}.$$

In the following, we focus on the case where $t$ becomes small.

We start from the exact analytical expression for $s_t(y, x)$ in Equation (21) given that $Q(y, x) > 0$, which is

$$s_t(y, x)$$

$$= \left( \sum_{u^{M \backslash j} \in [S]^{m(x)-1}} q_0(u^{M \backslash j}, y^j, x^{UM}) e^{-t(d-m(x))} \cdot e^{-t} \cdot q_{t|0}(x^{M \backslash j} | u^{M \backslash j}) \right) \Big/$$

$$\left( \sum_{a^j : a^j \neq [\mathrm{MASK}]} \sum_{u^{M \backslash j} \in [S]^{m(x)-1}} q_0(u^{M \backslash j}, a^j, x^{UM}) e^{-t(d-m(x))} (1 - e^{-t}) q_{t|0}(x^{M \backslash j} | u^{M \backslash j}) + \right.$$

$$\left. \sum_{u^{M \backslash j} \in [S]^{m(x)-1}} q_0(u^{M \backslash j}, [\mathrm{MASK}], x^{UM}) e^{-t(d-m(x))} q_{t|0}(x^{M \backslash j} | u^{M \backslash j}) \right)$$

$$\leq \left( \sum_{u^{M \backslash j} \in [S]^{m(x)-1}} q_0(u^{M \backslash j}, y^j, x^{UM}) e^{-t(d-m(x))} q_{t|0}(x^{M \backslash j} | u^{M \backslash j}) \right) \Big/$$

$$\left( \sum_{u^{M\backslash j} \in [S]^{m(x)-1}} q_0(u^{M\backslash j}, [\text{MASK}], x^{UM}) e^{-t(d-m(x))} q_{t|0}(x^{M\backslash j} | u^{M\backslash j}) \right)$$

$$= \left( \sum_{u^{M\backslash j} \in [S]^{m(x)-1}} q_0(y^j | u^{M\backslash j}, x^{UM}) q_0(u^{M\backslash j}, x^{UM}) e^{-t(d-m(x))} q_{t|0}(x^{M\backslash j} | u^{M\backslash j}) \right) \Bigg/$$

$$\left( \sum_{u^{M\backslash j} \in [S]^{m(x)-1}} q_0^j([\text{MASK}] | u^{M\backslash j}, x^{UM}) q_0(u^{M\backslash j}, x^{UM}) e^{-t(d-m(x))} q_{t|0}(x^{M\backslash j} | u^{M\backslash j}) \right)$$

$$\overset{(i)}{\leq} \left( \sum_{u^{M\backslash j} \in [S]^{m(x)-1}} q_0(y^j | u^{M\backslash j} q_0(u^{M\backslash j}, x^{UM}) e^{-t(d-m(x))} q_{t|0}(x^{M\backslash j} | u^{M\backslash j}) \right) \Bigg/$$

$$\left( \gamma \sum_{u^{M\backslash j} \in [S]^{m(x)-1}} \max_{a^j : a^j \neq [\text{MASK}]} q_0(a^j | u^{M\backslash j} q_0(u^{M\backslash j}, x^{UM}) e^{-t(d-m(x))} q_{t|0}(x^{M\backslash j} | u^{M\backslash j}) \right)$$

$$\leq \gamma^{-1} \tag{22}$$

where $(i)$ follows by Assumption 4. Note that this bound is uniform in $t$.

### I.7 Proof of Lemma 7

When $t$ is not so small, we can invoke Lemma 3 and get $\left| \frac{\partial}{\partial t} s_t(y, x) \right| \lesssim \frac{e^t}{(e^t-1)^2} \lesssim 1$. Thus, it suffices to get a non-diverging upper bound when $t$ becomes small.

Recall the proof of Lemma 3, in which we defined $T_1$ and $T_2$ such that $\frac{\partial}{\partial t} s_t(y, x) = T_1 - T_2$. For small $t$'s, we have $(1 - e^{-t}) \asymp t$ and $e^{-t} \asymp 1$. Also note that $m(x_0) \leq m_0$. Thus, for all such $t$'s, we can further simplify both terms as

$$T_1 = -\left( \sum_{u^{M\backslash j} \in [S]^{m(x)-1}} q_0(u^{M\backslash j}, y^j, x^{UM}) e^{-t} \cdot (1 - e^{-t})^{-m(u^{M\backslash j})} \frac{m(u^{M\backslash j}) + e^t - 1}{e^t - 1} \right) \Bigg/$$

$$\left( \sum_{a^j : a^j \neq [\text{MASK}]} \sum_{u^{M\backslash j} \in [S]^{m(x)-1}} q_0(u^{M\backslash j}, a^j, x^{UM})(1 - e^{-t})(1 - e^{-t})^{-m(u^{M\backslash j})} + \right.$$

$$\left. \sum_{u^{M\backslash j} \in [S]^{m(x)-1}} q_0(u^{M\backslash j}, [\text{MASK}], x^{UM})(1 - e^{-t})^{-m(u^{M\backslash j})} \right)$$

$$\asymp -\left( \sum_{u^{M\backslash j} \in [S]^{m(x)-1}} q_0(u^{M\backslash j}, y^j, x^{UM}) e^{-t} \cdot (1 - e^{-t})^{-m(u^{M\backslash j})} \frac{m(u^{M\backslash j}) + e^t - 1}{e^t - 1} \right) \Bigg/$$

$$\left( \sum_{u^{M\backslash j} \in [S]^{m(x)-1}} q_0(u^{M\backslash j}, [\text{MASK}], x^{UM})(1 - e^{-t})^{-m(u^{M\backslash j})} \right)$$

$$\asymp -\left( \sum_{u^{M\backslash j} : m(u^{M\backslash j}) = m_0} q_0(u^{M\backslash j}, y^j, x^{UM}) e^{-t} \cdot (1 - e^{-t})^{-m_0} \frac{m_0 + e^t - 1}{e^t - 1} \right) \Bigg/$$

$$\left( \sum_{u^{M\backslash j} : m(u^{M\backslash j}) = m_0} q_0(u^{M\backslash j}, [\text{MASK}], x^{UM})(1 - e^{-t})^{-m_0} \right)$$

$$= -\frac{m_0 + e^t - 1}{e^t - 1} \cdot \frac{\sum_{u^{M\backslash j} : m(u^{M\backslash j}) = m_0} q_0(u^{M\backslash j}, y^j, x^{UM})}{\sum_{u^{M\backslash j} : m(u^{M\backslash j}) = m_0} q_0(u^{M\backslash j}, [\text{MASK}], x^{UM})},$$

and

$$T_2 = -\left( \sum_{u^{M\backslash j} \in [S]^{m(x)-1}} q_0(u^{M\backslash j}, y^j, x^{UM}) \cdot e^{-t} \cdot (1 - e^{-t})^{-m(u^{M\backslash j})} \right).$$

$$\left( \sum_{a^j:a^j \neq [\text{MASK}]} \sum_{u^{M \backslash j} \in [S]^{m(x)-1}} q_0(u^{M \backslash j}, a^j, x^{UM})(m(u^{M \backslash j}) - 1)e^{-t}(1 - e^{-t})^{-m(u^{M \backslash j})} + \right.$$

$$\left. \sum_{u^{M \backslash j} \in [S]^{m(x)-1}} q_0(u^{M \backslash j}, [\text{MASK}], x^{UM}) \frac{m(u^{M \backslash j})e^{-t}}{1 - e^{-t}}(1 - e^{-t})^{-m(u^{M \backslash j})} \right) \Big/$$

$$\left( \sum_{a^j:a^j \neq [\text{MASK}]} \sum_{u^{M \backslash j} \in [S]^{m(x)-1}} q_0(u^{M \backslash j}, a^j, x^{UM})(1 - e^{-t})(1 - e^{-t})^{-m(u^{M \backslash j})} + \right.$$

$$\left. \sum_{u^{M \backslash j} \in [S]^{m(x)-1}} q_0(u^{M \backslash j}, [\text{MASK}], x^{UM})(1 - e^{-t})^{-m(u^{M \backslash j})} \right)^2$$

$$\asymp - \left( \sum_{u^{M \backslash j} \in [S]^{m(x)-1}} q_0(u^{M \backslash j}, y^j, x^{UM}) \cdot e^{-t} \cdot (1 - e^{-t})^{-m(u^{M \backslash j})} \right) \cdot$$

$$\left( \sum_{u^{M \backslash j} \in [S]^{m(x)-1}} q_0(u^{M \backslash j}, [\text{MASK}], x^{UM}) \frac{m(u^{M \backslash j})e^{-t}}{1 - e^{-t}}(1 - e^{-t})^{-m(u^{M \backslash j})} \right) \Big/$$

$$\left( \sum_{u^{M \backslash j} \in [S]^{m(x)-1}} q_0(u^{M \backslash j}, [\text{MASK}], x^{UM})(1 - e^{-t})^{-m(u^{M \backslash j})} \right)^2$$

$$\asymp - \left( \sum_{u^{M \backslash j}:m(u^{M \backslash j})=m_0} q_0(u^{M \backslash j}, y^j, x^{UM}) \cdot e^{-t} \cdot (1 - e^{-t})^{-m_0} \right) \cdot$$

$$\left( \sum_{u^{M \backslash j}:m(u^{M \backslash j})=m_0} q_0(u^{M \backslash j}, [\text{MASK}], x^{UM}) \frac{m_0 \cdot e^{-t}}{1 - e^{-t}}(1 - e^{-t})^{-m_0} \right) \Big/$$

$$\left( \sum_{u^{M \backslash j}:m(u^{M \backslash j})=m_0} q_0(u^{M \backslash j}, [\text{MASK}], x^{UM})(1 - e^{-t})^{-m_0} \right)^2$$

$$= -\frac{m_0}{e^t - 1} \cdot \frac{\sum_{u^{M \backslash j}:m(u^{M \backslash j})=m_0} q_0(u^{M \backslash j}, y^j, x^{UM})}{\sum_{u^{M \backslash j}:m(u^{M \backslash j})=m_0} q_0(u^{M \backslash j}, [\text{MASK}], x^{UM})}.$$

Therefore, when $t$ is small,

$$\left| \frac{\partial}{\partial t} s_t(y, x) \right| = |T_1 - T_2|$$

$$\asymp \left| \left( \frac{m_0}{e^t - 1} - \frac{m_0 + e^t - 1}{e^t - 1} \right) \frac{\sum_{u^{M \backslash j}:m(u^{M \backslash j})=m_0} q_0(u^{M \backslash j}, y^j, x^{UM})}{\sum_{u^{M \backslash j}:m(u^{M \backslash j})=m_0} q_0(u^{M \backslash j}, [\text{MASK}], x^{UM})} \right|$$

$$= \frac{\sum_{u^{M \backslash j}:m(u^{M \backslash j})=m_0} q_0(u^{M \backslash j}, y^j, x^{UM})}{\sum_{u^{M \backslash j}:m(u^{M \backslash j})=m_0} q_0(u^{M \backslash j}, [\text{MASK}], x^{UM})}$$

$$\leq \gamma^{-1}$$

where the last inequality follows similarly as in (22) when Assumption 4 holds. The proof is now complete.

