# OpenReview forum: "Absorb and Converge: Provable Convergence Guarantee for Absorbing Discrete Diffusion Models"
_NeurIPS.cc/2025/Conference — NeurIPS 2025 poster_

### Official Review · Reviewer_j85T · 2025-06-21

**Clarity:** 3
**Significance:** 3
**Originality:** 2
**Rating:** 4
**Confidence:** 4

**Summary:**

This paper studies the non-asymptotic convergence guarantees of absorbing discrete diffusion models, and establishes the algorithmic complexity of the $\tau$-leaping and uniformization samplers to obtain $\varepsilon$-accurate samples in KL divergence assuming $\varepsilon$-accurate score estimation. The paper also provides convergence guarantees without early-stopping under specific conditions on the target distribution.

**Questions:**

1. In Sec. 2, the score entropy loss is given in the form that includes the ground truth score $s _ t(y,x _ t)$, which is typically not available during training. As introduced in [Lou et al, ICML 2024], the form of the loss typically used in practice is the *denoising* score entropy, and I think the authors should clarify this point. Of course, in presenting the convergence bounds it is fine to use the score entropy loss there, but you should not say this is the "training objective".
2. The implementation of the $\tau$-leaping discussed in Sec. 2 lacks the **clipping** step. If many jumps happen in a certain position, then summing up the difference may make the value out of the domain $[S]$. In practical implementations one typically truncates the value to the domain $[S]$ or just rejects the jump if there are more than one jump in a certain position (which happens w.p. $o(h)$ where $h$ is the step size). See [Ren et al, ICLR 2025, Rmk. A.13] for a discussion of this issue. I have not seen how the authors handle this issue in the paper, and a brief clarification would be helpful.
3. Assump. 4 is non-standard. As far as I know, in almost all of the real-world implementations, the mask token is different from any of the tokens appearing in the dataset, i.e., the mask token is not in the support of the data distribution $q _ 0$. But in Assump. 4, the authors assume the mask token is one of the existing tokens in the dataset and takes a conditional probability of at least $\gamma$, which, though not impossible, is not the common case, and $\gamma$ would be extremely small in high-dimensional distributions with large vocabulary size.
4. Lem. 2 seems to be strange. From Lem. 1 and [Ou et al., ICLR 2025], we know the score is just a time-dependent scalar (here $\frac{\mathrm{e}^{-t}}{1-\mathrm{e}^{-t}}$) times the conditional probability of $y^j$ given all the unmasked positions in $x$ in the data distribution $q _ 0$. But why can the conditional probability be lower bounded in general? Is there any hidden assumption or am I missing something?
5. To the best of my knowledge, the first use of the "Jensen approach" for continuous diffusion models is in [Chen et al, ICML 2023, Lemma C.4] (reference [40] in this paper), not the reference given in the paper.

**Ethical Concerns:**

["NO or VERY MINOR ethics concerns only"]

**Final Justification:**

During rebuttal, the authors have addressed most of my concerns, especially for the motivation of this research (which we had a long discussion). I thus raise the rating to 4 to reflect the authors' commitment of improving the paper and effort in better clarifying the value.

**Limitations:**

I do not see clear discussion of limitations in the paper though the authors claimed having done so in the checklist. See the weakness part above for my opinions.

**Quality:**

2

**Strengths And Weaknesses:**

**Strengths:**

The paper provides the first (to the best of my knowledge) non-asymptotic analysis of the convergence guarantees for the absorbing discrete diffusion models, complementing the theoretical understanding of discrete diffusion models that have been mainly studied in the uniform setting. The paper is clearly written and all the main results are well presented.

**Weaknesses:**

I have the following concerns about the motivation of the paper, and they impact my willingness to recommend acceptance of this work.

The paper studies two algorithms for sampling from absorbing discrete diffusion models: $\tau$-leaping and uniformization. While these are useful samplers for uniform discrete diffusion models, they are typically not used very often in the setting of absorbing discrete diffusion models. As studied in [Ou et al., ICLR 2025] and several other works such as [Sahoo et al., NeurIPS 2024], the absorbing discrete diffusion models can be *exactly* sampled via random-order autoregressive sampling in *exactly* $d$ steps. Exact sampling means that there is no discretization error (like in the $\tau$-leaping method) and no initialization error as discussed in Thm. 1 of your paper, and the KL divergence from the target distribution to the sampled distribution is bounded above by the score learning error. This is a significant advantage over the $\tau$-leaping and uniformization methods that are studied in this paper, which require more than $d$ steps to sample from as indicated by your bounds.

In light of this, I feel it is not of high value and significance to the community (both theoretical and applicational) to study the convergence guarantee of these sampling algorithms that may contain numerical error and require significantly more steps, unless there is a strong motivation to do so, such as being able to sample in $\widetilde{O}(\sqrt{d})$ steps (see, e.g., [Ren et al., 2025](https://arxiv.org/abs/2502.00234)), or to prove some other theoretical results for inexact sampling with better generation quality.

Besides, there are also some issues regarding the special structure of the absorbing discrete diffusion models that can be used to make the analysis easier. For instance, as the time parameter is redundant in the absorbing discrete diffusion models (proved in [Ou et al., ICLR 2025] and several other papers), Thm. 1 is totally unnecessary with a different noise schedule: one can choose a function $\sigma(t)$ and write $Q _ t=\sigma(t)Q^\mathrm{base}$. If $\int _ 0^T\sigma(t)\mathrm{d}t=\infty$, then the distribution at the final time $T$ is exactly $(\delta _ {[\mathrm{MASK}]})^{\otimes d}$. Also, in Lem. 1, $q^j _ {0|t}$ can be written as $q^j _ 0(y^j|x^\mathrm{unmasked})$, which is a more natural way to write the conditional distribution of the unmasked variables given the masked variables.

---

> ### Author Rebuttal · Authors · 2025-07-30
>
> We thank the reviewer for carefully reading our work and providing insightful feedback.
>
> **Q1:** Weakness and question on the value and significance of studying $\tau$-leaping and uniformization algorithms in the setting of absorbing discrete diffusion models.
>
> **A1:** We respectfully disagree with the reviewer. The references cited by the reviewer do not appear to support the claim that the algorithms studied in our paper are not used very often in the setting of absorbing discrete diffusion models. In practice, $\tau$-leaping-like samplers—such as the Euler method and Tweedie $\tau$-leaping—have been employed in various works under absorbing settings (e.g., [3–4]), including the references [1] and [2] provided by the reviewer. Our reading of [1] and [2] suggests that sampling in those works is not performed in exactly $d$ steps, as asserted. In both papers’ discrete diffusion models, during each sampling step, there is a non-zero probability for a mask state to remain as mask, yielding the total number of steps (much) greater than $d$ (i.e., not “exact”). Such non-“exact-step” sampling is also indicated by their empirical experiments as well. For example, in [1, Table 3], the number of sampling steps reaches up to 4096, exceeding the window length of $d = 1024$. Similarly, in [2, Table 6], with a window size of $d = 1024$ tokens, the number of steps can go up to 10,000.
>
> **Comparison between exact samplers and $\tau$-leaping-like samplers:** Since exact samplers are often in autoregressive-type masked language models, we refer to it as **MLM** in our following discussion, and use **DDM** to denote the discrete diffusion model with $\tau$-leaping type samplers. While MLMs offer certain advantages such as no discretization and initialization errors and a deterministic number of sampling steps (e.g., $d$ steps), there are several fundamental reasons why they may still underperform compared to DDMs in certain settings. (1) DDMs sample tokens iteratively based on a learned reverse process that models dependencies across all tokens and time steps. In contrast, MLMs often rely on shallow, single-step predictions for each masked token, limiting their ability to model long-range or high-order dependencies in a globally consistent manner. For example, [11] shows that DDMs can model structured sequences more effectively than autoregressive or MLM-type methods, due to the ability to revise and refine generations. (2) The deterministic $d$-step schedule in MLMs may not optimally align with the complexity of the generation task. DDMs allow adaptive generation paths, e.g., more steps for difficult tokens, enabling finer-grained control and potentially better outputs. [12] reveals that MLMs allocate separate representational dimensions to [MASK] tokens, limiting downstream expressiveness and overall performance.
>
> Overall, the current literature continues to actively explore and compare exact samplers in MLMs and $\tau$-leaping-like samplers in DDMs. As far as we can tell, there is no conclusive evidence that one method consistently outperforms the other (e.g., [10, Table 3]). Therefore, we believe both sampling techniques offer valuable and complementary perspectives, and hence both warrant continued theoretical and empirical investigation.
>
> **Our another motivation:** We further note that another motivation for us is to theoretically understand and justify the benefits of using absorbing rate over uniform rate, as empirically observed in [3]. To this end, we choose the same set of samplers ($\tau$-leaping and uniformization) as those typically used for the uniform rate. Overall, we show improved rates of these samplers on absorbing rates over uniform-rate counterparts.
>
> **A side note regarding [6]:** [6] does not seem to have an explicit result that characterizes the dependence of $d$. It is unclear whether the sampling is in terms of $\tilde{O}(\sqrt{d})$ steps.
>
> **Q2.1:** Question on the noise schedule.
>
> **A2.1:** In our study, we follow many prior works (under the uniform rate) in assuming time-homogeneity (e.g., [7-9], which study uniform discrete diffusion models). This allows for a more direct comparison on the rate matrices by controlling for other parameter differences, such as the noise schedule. While it might be helpful to introduce a non-homogeneous noise schedule to tackle the initialization error, both the discretization error and the estimation error will be affected, which may possibly worsen the overall convergence rate.
>
> **Q2.2:** Question on notation in Lem. 1.
>
> **A2.2:** To clarify, our $q_{0|t}^j(y^j|x)$ in Lemma 1 is **different** from the “clean-data distribution” as in [1, Theorem 1]. While their “clean-data distribution” does not depend on $t$, ours is the **posterior** distribution of $x_0^j=y^j$ given that $x_t = x$ (notably, at time $t$), which obviously depends on $t$.
>
> Now we argue that $q_{0|t}^j(y^j|x)$ must depend on $t$ in general in the following example. Suppose that the score function is well-defined at $t=0$ (i.e., finite for every pair of x and y having Hamming distance 1). (Necessarily, this implies that [MASK] exists in the training data.) Then, since the time-dependent coefficient $e^{-t}/(1-e^{-t})$ blows up as $t \to 0$, the other term in the multiplication must also depend on $t$ so that the overall product remains a finite quantity.
>
> **Q3:** Question on score entropy loss.
>
> **A3:** We will clarify this point in the revised draft. In particular, we will change the previous one to “training loss” and add the following sentence: “In practice, for tractable training, the denoising score entropy is usually used, which is a variant of the score entropy [9].”
>
> **Q4:** Question on the lacking of clipping step in $\tau$-leaping discussed in Sec. 2.
>
> **A4:** Indeed, in practice, an additional clipping step is necessary to avoid boundary crossing behaviors. As shown in [7], such a step does not affect the convergence rate given sufficiently small step-sizes (cf. [7, Remark A.13]). We will add this piece of clarification in the footnote when introducing the sampler.
>
> **Q5:** Question on Assump. 4.
>
> **A5:** To clarify, our main theoretical guarantees with early-stopping (i.e., Theorems 2–3) do not rely on Assumption 4 and thus remain broadly applicable. Early-stopping is often used in theoretical analysis to deal with practical difficulty. Thus, in practice, one can always rely on early-stopping for a small enough initial time $\delta$.
>
> Our dependence on Assump. 4 comes only if we aim to obtain stronger theoretical guarantees to remove early stopping (which is generally regarded as very difficult and has been very rarely established in the literature). In fact, Assumption 4 can be justified as nearly necessary for the validity of **the diffusion algorithm itself**, as follows. By the second part of Lemma 2, if Assumption 4 is violated, the score function will (nearly) diverge around $t = 0$. Since the diffusion algorithm relies on the score function to make progress at each step, such divergence at $t = 0$ would render the algorithm itself invalid in that regime. We will make this point clear in our revised version.
>
> **Q6:** Question on Lem. 2.
>
> **A6:** As we pointed out in A2.2, our $q_{0|t}^j(y^j|x)$ in our Lemma 1 is not the “clean-data” distribution but the posterior distribution which depends on $t$. Meanwhile, our Lemma 1 is more general than [1, Theorem 1] by taking into account the possibility of masks in the data.
>
> In Lemma 2, instead of providing a lower bound for this posterior distribution, we provide the bound **directly for the score function** through Bayes’ rule. Intuitively, if $q_t$ has full support for all $t > 0$ (necessarily, this also holds for $q_0$ due to continuity of measure), all possible states (including the mask) would have finite probability under $q_0$, which implies a finite lower bound for the score function. Otherwise, the score function will diverge, at the same rate of the time-dependent coefficient.
>
> **Q7:** Reference related to "Jensen approach".
>
> **A7:** Thank you for pointing this out! We will change the reference and further clarify this in the proof in Appendix D.
>
> We thank the reviewer once again for your thoughtful comments. We hope our responses have addressed your concerns. In particular, we hope the reviewer could kindly consider the value of exploring absorbing diffusion models with stochastic sampling, which is still a meaningful and timely direction of study. If so, we would greatly appreciate it if you could kindly reconsider your evaluation and consider raising the score. We would be happy to address any further questions or clarifications you may have.
>
> [1] Ou, Jingyang, et al. “Your Absorbing Discrete Diffusion Secretly Models the Conditional Distributions of Clean Data.”
>
> [2] Sahoo, Subham Sekhar et al. “Simple and Effective Masked Diffusion Language Models.”
>
> [3] Lou, Aaron, et al. “Discrete diffusion modeling by estimating the ratios of the data distribution.”
>
> [4] Nisonoff, Hunter, et al. “Unlocking Guidance for Discrete State-Space Diffusion and Flow Models.”
>
> [5] Ghazvininejad, Marjan et al. “Mask-Predict: Parallel Decoding of Conditional Masked Language Models.”
>
> [6] Ren, Yinuo, et al. "Fast solvers for discrete diffusion models: Theory and applications of high-order algorithms."
>
> [7] Ren, Yinuo, et al. “How Discrete and Continuous Diffusion Meet: Comprehensive Analysis of Discrete Diffusion Models via a Stochastic Integral Framework.”
>
> [8] Zhang, Zikun, et al. “Convergence of Score-Based Discrete Diffusion Models: A Discrete-Time Analysis.”
>
> [9] Chen, Hongrui and Ying, Lexing. “Convergence analysis of discrete diffusion model: Exact
> implementation through uniformization.”
>
> [10] Shi, Jiaxin, et al. “Simplified and Generalized Masked Diffusion for Discrete Data.”
>
> [11] Austin et al. “Structured denoising diffusion models in discrete state-spaces”
>
> [12] Meng et al. “Representation deficiency in masked language modeling”

---

> > ### Comment · Reviewer_j85T · 2025-08-04
> >
> > Reviewer response
> >
> > I thank the authors for their detailed response to my comments.
> >
> > **A1:** For the masked discrete diffusion model, we all agree that each position only jumps once from mask to some unmasked position. Thus, when the sampling method does not involve any remasking step, you should be able to draw a sample in NFE $\le$ the sequence length. NFE $>$ sequence length is redundant because there is no change in the samples at some steps so you could reuse the cached score network output. For instance, [13] proposed a first-hitting sampler to avoid redundant NFEs. As your bounds are much larger than $O(d)$, I believe you can improve it to exactly $d$ with a similar caching and first-hitting approach without changing the sampling trajectory.
> >
> > **Comparison between exact samplers and $\tau$-leaping-like samplers**: Thank you for pointing out the reference. This is an interesting point, but I can't completely agree. DDMs don't learn the unmasking order of all tokens. In $\tau$-leaping or uniformization, conditional on the current state, the next position to unmask can be proved to be uniform from the remaining masked positions. The reference [11] you cited has a refining step, which is not involved in the samplers you analyzed. Similarly, the adaptive generation path does not fall into any of the sampling methods (exact, $\tau$-leaping, or uniformization). Of course the study of samplers in masked diffusion models is an interesting ongoing research direction, but I don't think the current analysis on $\tau$-leaping and uniformization is of much help: (i) they can be implemented in a more efficient way in terms of NFE without discretization error, and (ii) your paper does not show evidence that $\tau$-leaping or uniformization is better in terms of sampling quality (e.g., generalization properties, diversity, etc.). I agree with another motivation you mentioned in comparing with uniform discrete diffusion models, which shows the good properties of the masked diffusion models.
> >
> > For [6], I apologize for not carefully reading that paper. The second order convergence does not directly indicate $\widetilde{O}(\sqrt{d})$ NFE.
> >
> > **A2.1**: Thanks for the clarification. I don't think non-homogeneous noise schedule would worsen the overall convergence rate. I hope the authors could try it in the future, but it's OK to use it in the current analysis.
> >
> > **A2.2 & A6**: Thanks for the clarification. As I pointed out in the initial review, in almost all of the implementations, the mask is a separate token not in the data distribution. This is not a good assumption and caused lots of confusion to me. I would suggest the authors to clarify this early in the paper, and not use this assumption in the analysis.
> >
> > Regarding rating, in general I don't recommend acceptance for this paper due to its limitations, but I'll increase it to 3 to reflect the authors' efforts in addressing my comments.
> >
> > **Reference:** [13] Zheng et al. Masked Diffusion Models are Secretly Time-Agnostic Masked Models and Exploit Inaccurate Categorical Sampling. ICLR 2025.

---

> > > ### Author Response · Authors · 2025-08-05
> > >
> > > We thank the reviewer for the thoughtful feedback!
> > >
> > > **Regarding NFE:** Thanks for clarifying this point, Indeed, it is easy to see that our $\tau$-leaping sampler also achieves $d$ NFE when adopting the cached strategy used in [1]. Our current characterization uses the conventional metric of the total number of sampling steps, including those with and without state changes. We will clarify this point in the revision.
> > >
> > > **Regarding [13]:** Note that the main idea of [13] is very similar to the uniformization sampler that we analyze, where the next sampling time is randomly drawn before sampling for the next state. Although the uniformization sampler would first discretize the sampling path, the number of transitions within each discretization is a Poisson random variable (which might be zero, i.e., no function evaluation during the given period). Indeed, our Theorem 3 shows (i) that the uniformization sampler does not have discretization error, and (ii) that the expected number of state transitions (i.e., NFE) is $O(d)$ (disregarding log quantities). These two properties align with the expected properties of the sampler in [13], as noted by the reviewer. While there are some subtle differences between the two algorithms, our analysis can serve as a steppingstone for further research on this type of samplers.
> > >
> > > **Regarding $\tau$-leaping like samplers:** While those more refined samplers such as the Euler method and Tweedie $\tau$-leaping incorporate additional features beyond the simplified version we analyze, our current proof framework can still be useful for several key proof components. For example, the theoretical characterizations of the absorbing score functions (Lemmas 1 and 2) can be useful for any absorbing-type diffusion models that use concrete scores. Also, the discretization error for these algorithms can be decomposed in a similar way as our Theorem 2 to obtain the convergence rate. Hence, this paper serves as a starting point and building block for analyzing increasingly sophisticated sampling strategies for the masked diffusion model.
> > >
> > > **Regarding separate token not in the data distribution:** Thank you for the suggestion. In the revised version, we will clarify that Assumption 4 may not align with practical masking strategies. We are also happy to remove the part associated with Assumption 4, or move it to the supplementary materials if the reviewer feels strongly that it should be excluded. That said, we feel that the study under this assumption still offers theoretical value in understanding whether early stopping is necessary for guaranteed convergence. Specifically: (1) Without Assumption 4, establishing provable convergence without early stopping is nearly impossible, as argued in our previous response; and (2) With Assumption 4, the current samplers without early stopping can be shown to converge provably. Since the role of early stopping is an interesting theoretical aspect, we felt that including the study under this assumption helps to clarify when and why it may be necessary.
> > >
> > > We will also clarify that our $q_{0|t}^j$ in Lemma 1 is different from the “clean-data distribution” as in [1, Theorem 1].
> > >
> > > We thank the reviewer again for your time and efforts. We hope that our responses have addressed your further concerns. If so, we would be sincerely grateful if you could consider updating your rating to reflect that. Of course, we are more than happy to address any further questions you may have.

---

> > > > ### Comment · Reviewer_j85T · 2025-08-07
> > > >
> > > > I sincerely thank the authors for clarifying my questions.
> > > >
> > > > **NFE**: I now understand that you are measuring the total number of sampling steps, while the NFE can be reduced to $O(d)$ due to caching. For the analysis, NFE is a more important metric compared to the total number of steps, as querying the model costs more computation time, especially for huge models. Hence, NFE, rather than the total number of steps, should be the main metric for comparison in the paper. In terms of NFE, the current analysis does not provide a better bound than $O(d)$ unless you allow large sampling errors.
> > > >
> > > > **First-hitting v.s. uniformization**: Thanks for making the comparison. Both of them, together with random order AR sampling, are *exact* samplers in terms of zero discretization error of the CTMC sampling path. As they share the same nature, and FHS requires less NFEs than uniformization (without caching), this makes FHS a better choice.
> > > >
> > > > **Separate token not in the data distribution**: Thank you for the explanation of the assumption and its theoretical implications. I now don't see it as a major issue, but please clarify it in the revision. For the early stopping, I'd still like to point out that this is due to *your choice of sampler*, early stopping can be totally avoided by not using $\tau$-leaping or the version of uniformization in the paper.
> > > >
> > > > Finally, my main concern is still on whether it is of significant theoretical value to analyze these two samplers in the paper:
> > > >
> > > > - For $\tau$-leaping, it involves discretization error, while a cached version can only reduce NFE to $O(d)$ without sacrificing the sampling quality too much.
> > > > - For uniformization, there are equivalent exact samplers (random order AR, FHS) that require provably less steps and less NFEs (when caching is not used) or the same $O(d)$ NFEs (when caching is used).
> > > >
> > > > I agree that technically the analysis provides some insights and contributes to the community, but I am not fully convinced about the value.

---

> > > > > ### Author Response · Authors · 2025-08-07
> > > > >
> > > > > We thank the reviewer very much for your insightful further comments. We hope that our response below could further convince you of the value of the current work.
> > > > >
> > > > > **NFE and the number of steps:** While we totally agree that NFE is an important metric to consider given its close relationship to the amount of computational resources needed, it is not clear whether it is more important than the number of steps. To be sure, especially for deterministic-step samplers (which are still quite prevalent [1,2,10]), even if the number of state changes (i.e., NFE) is exactly $d$, the number of sampling still increases with a larger number of steps. In other words, even if the sampled state remains at some step (such that the cached score can be used), there is still a need to actually carry out the sampling process, resulting in a higher computation cost due to sampling. The number of steps is also a conventional metric in **almost all theoretical literature** on the convergence analysis of diffusion models (e.g., [6-9]). Therefore, the number of steps is still of value to consider. We fully recognize the importance of NFE and consider improving it to be a valuable and promising direction for future work.
> > > > >
> > > > > **Regarding samplers:** While we agree with the reviewer’s comments regarding FHS and uniformization about zero discretization error and NFE comparison, we also note that the final performance is influenced not only by discretization error and NFE, but also, for example, by **estimation error** (i.e., the inaccuracy in the score function estimation in order to construct the posterior sampling probabilities), which can play a significant role. Because of this, it is not clear that FHS is consistently better than uniformization, or they are consistently better than other deterministic samplers, especially when the system scenario is away from the ideal situation. For example, when the NFE becomes lower, FHS may require a smaller score estimation error to maintain the same level of sample quality. As a result, a slightly increased score estimation error (that is possibly due to computational budgets) can, in turn, degrade the final sample quality. In comparison, the dependence on the estimation error is explicit and low for the two samplers selected through our analysis: $O(1/\epsilon)$ for $\tau$-leaping and $O(\log\log(1/\epsilon))$ for uniformization.
> > > > >
> > > > > **Regarding early-stopping:** To clarify, the need for early-stopping is not because of the choice of our sampler. Rather, it is **built-in in the problem setting**. To see this, our Lemma 2 specifies a **diverging** lower bound on the concrete score function (cf. [1, Theorem 1]). This means that if there is no mask in the data distribution, the diffusion reverse process is not properly defined when $t \to 0$. Note that here it does not matter which algorithm we use to sample from the reverse process. To this end, our Assumption 4 specifies a nearly necessary condition under which the reverse process is still well-defined (and thus early-stopping is not needed).
> > > > >
> > > > > We will definitely clarify Assumption 4 in our revision as you have suggested.
> > > > >
> > > > > **Regarding theoretical value to analyze $\tau$-leaping and uniformization:** We would like to highlight three key aspects: **(1)** As we noted in our previous response, which the reviewer has kindly acknowledged, both samplers have been extensively studied in the context of uniform-rate discrete diffusion models. Studying these samplers in the absorbing-rate setting provides a meaningful comparison between the two settings, which will be of theoretical interest to the theoretical research community. **(2)** These samplers remain actively being improved in practical applications. As the first theoretical convergence analysis of these samplers for the absorbing rate (to the best of our knowledge), our work lays the foundation and is a necessary step toward enabling future analysis of more sophisticated or improved variants. **(3)** Regardless of which type of samplers may ultimately be preferred in practice, our theoretical study of these samplers provides provable insights into their relative strengths and limitations.
> > > > >
> > > > > We sincerely thank the reviewer again for your continued engagement in this productive and insightful discussion. We hope our responses have addressed your concerns. If so, we would be grateful if you would consider raising your score to reflect that. Of course, we would be happy to answer any further questions you may have.

---

> > > > > > ### Comment · Reviewer_j85T · 2025-08-08
> > > > > >
> > > > > > I thank the authors for their detailed and thoughtful responses to my comments.
> > > > > >
> > > > > > **NFE**: Thanks for the clarification. I also carefully read through the paper [13] and found the paragraphs "Categorical Sampling is Time-Consuming" and "Caching Strategy Degrades in Batched Sampling" may be more relevant and convincing for answering my concerns, which explains why the number of steps may be a more critical metric for evaluating the computational cost for masked diffusion models even with caching strategy. Therefore, I agree with you on the use of the number of steps.
> > > > > >
> > > > > > **Samplers**: As I have mentioned before and we all agree, the only error in FHS and uniformization is the score learning error. And in fact, you can actually prove that the path measures are equivalent in the two settings, i.e., they would finally output the same distribution, which is the final-time distribution of the CTMC given by the inaccurately learned generator. Thus I believe theoretically FHS and uniformization have equal performance. Also if I understand correctly, FHS as proposed in [13] needs NFE exactly $d$ -- there is no way of using smaller NFE to simulate FHS.
> > > > > >
> > > > > > **Early-stopping**: I agree the unboundedness of the score when $t\to0$, but when there is no mask in the data distribution, the score is just a scalar depending on time (that goes to infinity when $t\to0$) multiplied by the conditional probability in the clean data, and thus you can always remove the time-dependent scalar and use the conditional probability to sample -- this does not cause numerical issues, and I believe the diffusion reverse process is still properly defined.
> > > > > >
> > > > > > Regarding other points including the theoretical value, thank you again for the clarification. I'm glad we are gradually making progress in better understanding the value of this paper, though there may be still some minor concerns. I'll carefully consider about increasing the rating to 4.

---

> > > > > > > ### Author Response · Authors · 2025-08-08
> > > > > > >
> > > > > > > We thank the reviewer very much for your tremendous efforts in engaging with us and for providing such insightful feedback.
> > > > > > >
> > > > > > > We are pleased that we finally reached a consensus on the value of analyzing the number of steps. For samplers, we agree with your comments that (the first-order token-by-token) FHS and uniformization output the same path measure, which corresponds to the estimated continuous reverse process. For early-stopping, we fully agree that your algorithm can avoid numerical issues. We also want to quickly note that it is in theory different from the diffusion reverse process, which depends on the true score function (that includes the time-dependent scalar) and thus can be considered not yet fully characterized. That said, your proposed algorithm is an interesting future direction to consider for removing early-stopping.
> > > > > > >
> > > > > > > We will carefully revise our paper based on all your comments and suggestions during this review process. Thank you again for providing such thoughtful feedback.
> > > > > > >
> > > > > > > We sincerely thank the reviewer for the willingness to carefully consider increasing the score, and we hope this will result in a positive final decision. Again, we truly appreciate your time and effort, and we greatly enjoyed our productive discussion throughout the review process.

---

> > > > > > > > ### Comment · Reviewer_j85T · 2025-08-09
> > > > > > > >
> > > > > > > > Thank you for the reply, which has helped better clarify the value of this work. I will raise the rating to 4.

---

> > > > > > > > > ### Author Response · Authors · 2025-08-09
> > > > > > > > >
> > > > > > > > > We sincerely thank the reviewer for the dedicated efforts and for raising the rating. We greatly appreciate your constructive suggestions and insightful comments throughout this review process.

---

> ### Author Response · Authors · 2025-08-04
>
> Dear Reviewer j85T,
>
> As the author-reviewer discussion period will end soon, we would like to check whether our responses have properly addressed your concerns? If so, could you please kindly consider increasing your initial score accordingly? Certainly, we are more than happy to answer your further questions.
>
> Thank you for your time and effort in reviewing our work!
>
> Best Regards,
> Authors

---

### Official Review · Reviewer_vZMc · 2025-06-28

**Clarity:** 3
**Significance:** 3
**Originality:** 2
**Rating:** 5
**Confidence:** 1

**Summary:**

This paper studies the discrere diffusion models with absorbing rate matrices. The authors establish the first convergence guarantees for both the τ-leaping and uniformization samplers under absorbing rate matrices, which are better or same for uniform rate matrix. Thses results are based on several new techniques.

**Questions:**

A more detailed discussion of the dependence on M, along with a comparison to existing literature, would be appreciated.

**Ethical Concerns:**

["NO or VERY MINOR ethics concerns only"]

**Final Justification:**

My concerns and questions have been addressed, and although I am not an expert in discrete diffusion models, I believe this work is worthy of acceptance.

**Limitations:**

yes

**Quality:**

3

**Strengths And Weaknesses:**

Strength:

This paper delivers the first rigorous convergence analysis for absorbing discrete diffusion models and introduces several technically significant innovations. By proposing a surrogate initialization distribution, it effectively overcomes the challenge of ill-defined KL divergence caused by a singleton stationary state. The authors further employ novel Jensen-type arguments and refined score-control techniques, thereby eliminating reliance on log-Sobolev inequalities. Importantly, the work establishes convergence guarantees under both early-stopping and non-early-stopping regimes, offering a comprehensive theoretical foundation.


Weakness:

The assumption of bounded score estimate seems too strong. Could you give more justification?

---

> ### Author Rebuttal · Authors · 2025-07-30
>
> We thank the reviewer for carefully reading our work and providing encouraging and insightful feedback.
>
> **Q1:** The assumption of the bounded score estimate seems too strong. Could you give more justification?
>
> **A1:** Definitely! Assumption 2 is commonly adopted in the previous studies for uniform-rate discrete diffusion models (e.g., [15] and [16]). In practice, this can be satisfied with score-clipping during training [16]. Indeed, the convergence error bounds in our main results only at most depend on $\log M$. In the revision, we will add these sentences after introducing Assumption 2.
>
> **Q2:** A more detailed discussion of the dependence on M, along with a comparison to existing literature, would be appreciated.
>
> **A2:** We thank the reviewer for raising this. As follows we divide the existing literature according to two types of sampling algorithms. First, for random-stepsize samplers, the uniformization algorithm in our paper does not depend on $M$ under the absorbing rate (with or without early-stopping). This result aligns with that under the uniform rate in [15]. For deterministic-stepsize samplers, the dependence on $M$ was initially shown to be $O(\sqrt{M})$ in [16] and was later improved to be $O(\log M)$ [15], both under the uniform rate. In comparison, our result has shown the same $O(\log M)$ dependence under the absorbing rate.
>
> We sincerely appreciate the reviewer’s continued engagement. We hope our responses have fully addressed the concerns. We are also happy to provide further clarification if needed.

---

> > ### Comment · Reviewer_vZMc · 2025-08-06
> >
> > Thank you for your response. I will maintain my current score in support of this work.

---

### Official Review · Reviewer_rP18 · 2025-06-29

**Clarity:** 4
**Significance:** 3
**Originality:** 4
**Rating:** 5
**Confidence:** 3

**Summary:**

This paper presents the first provable convergence guarantees and error analysis for discrete diffusion models utilizing absorbing rate matrices. It addresses a critical gap, as prior theoretical work mostly focused on uniform rate matrices despite absorbing models showing superior empirical performance. The authors derive finite-time error bounds and analyze convergence rates for $\tau$-leaping and uniformization samplers, demonstrating improved rates. Notably, they achieve convergence without early-stopping under certain conditions and introduce novel technical tools for bounding absorbing score functions. The theoretical insights reinforce the observed advantages of absorbing discrete diffusion models.

**Questions:**

1. Regarding Assumption 4 (for convergence without early-stopping), could you further elaborate on its practical implications and the types of data distributions for which it is most likely to hold? Specifically, how might practitioners determine if their specific setup satisfies this assumption, or what might be the consequences if it does not?

**Ethical Concerns:**

["NO or VERY MINOR ethics concerns only"]

**Final Justification:**

This is a very nice work on the fundamental theory of discrete diffusion model. I will keep my rate to accpet this work.

**Limitations:**

yes

**Paper Formatting Concerns:**

yes

**Quality:**

4

**Strengths And Weaknesses:**

- Strengths

1. Good Writing: The paper is well-written and structured, making the complex theoretical arguments as accessible as possible. The authors adeptly highlight their key proof techniques, which significantly aids in understanding the methodological contributions.

2. Theoretical Contributions: This paper delivers the first rigorous theoretical analysis of sampling convergence for discrete diffusion models that employ absorbing rate matrices. This fills a significant gap in the literature and provides a foundational understanding for models whose empirical performance has often surpassed those using uniform rate matrices.

3. Explanation on Empirical Results: The authors provide a clear theoretical explanation for why absorbing rate matrices offer advantages over uniform rate matrices in practice. Their analysis includes improved convergence rates for common samplers, such as $\tau$-leaping, aligning theory with observed empirical benefits.

- Weaknesses

1. Practical Concern on Assumption 4: My primary concern lies with Assumption 4, used to establish convergence without early-stopping. While the paper acknowledges that previous works might make stronger assumptions, this particular condition still appears quite strict for practical distributions. It also seems that current absorbing discrete diffusion models do not typically rely on such an assumption when determining mask tokens, which might limit its immediate practical applicability.

---

> ### Author Rebuttal · Authors · 2025-07-30
>
> We thank the reviewer for carefully reading our work and providing encouraging and insightful feedback.
>
> **Q1:** Practical Concern on Assumption 4: My primary concern lies with Assumption 4, used to establish convergence without early-stopping. While the paper acknowledges that previous works might make stronger assumptions, this particular condition still appears quite strict for practical distributions. It also seems that current absorbing discrete diffusion models do not typically rely on such an assumption when determining mask tokens, which might limit its immediate practical applicability.
>
> **A1:** Even though it seems strict, Assumption 4 can be justified as nearly necessary for the validity of **the diffusion algorithm itself** (without early stopping), as follows. By the second part of Lemma 2, if Assumption 4 is violated, the score function will (nearly) diverge around $t = 0$. Since the diffusion algorithm relies on the score function to make progress at each step, such divergence at $t = 0$ would render the algorithm itself invalid in that regime. We have made this point clear after Assumption 4 in our revised draft.
>
> **Q2:** Regarding Assumption 4 (for convergence without early-stopping), could you further elaborate on its practical implications and the types of data distributions for which it is most likely to hold? Specifically, how might practitioners determine if their specific setup satisfies this assumption, or what might be the consequences if it does not?
>
> **A2:** Excellent question. Note  that our main theoretical guarantees with early-stopping (i.e., Theorems 2–3) do not rely on Assumption 4 and thus remain broadly applicable. Thus, practitioners can always rely on early stopping for a small enough initial time $\delta$, which would make the approach practical. Specifically for Assumption 4, it can be satisfied when the probability of [MASK] is non-zero in the data. Also, as argued above, given that Assumption 4 is nearly necessary for the diffusion algorithm to be valid, the violation of it stops one from pushing too much towards $t=0$.
>
> We thank the reviewer again for your comments. We hope that our responses resolved your concerns. Certainly, we are more than happy to answer your further questions if any.

---

> > ### Comment · Reviewer_rP18 · 2025-08-05
> >
> > Your explanation could help me have a better understanding of your result.I have no further questions.

---

### Official Review · Reviewer_gMRL · 2025-07-03

**Clarity:** 3
**Significance:** 2
**Originality:** 3
**Rating:** 4
**Confidence:** 2

**Summary:**

This paper provides convergence rate analysis for discrete diffusion models using absorbing rate matrices.  The authors came up with a new way to handle the singleton distribution caused by the absorbing state. They also show that two popular sampling methods—τ-leaping and uniformization—work faster and better with absorbing rates than with uniform rates. They also find conditions where early stopping isn’t needed. Overall, this work shows that using the absorbing rate matrix can achieve convergence with fewer sampling steps—scaling linearly with the data dimension instead of quadratically

**Questions:**

There are some divergences which remain well-defined when the underlying distribution is a singleton or non a.c., e.g. the spread KL divergence [1].Do you think it is possible to use such divergence to get rid of the surrogate distribution?


[1] [Spread Divegrence](https://arxiv.org/abs/1811.08968)

**Ethical Concerns:**

["NO or VERY MINOR ethics concerns only"]

**Final Justification:**

I  am not an expert in the discrete diffusion theory and I do not have concerns left.

**Limitations:**

While the theoretical contributions are strong, the paper currently lacks empirical verification or practical demonstrations showing how the proposed convergence bounds translate to real-world implementations. For instance, it would be valuable to see whether the improved convergence rates under absorbing rate matrices indeed yield faster or higher-quality sampling in practice. Some plots would be useful to support the theoretical argument.

**Quality:**

3

**Strengths And Weaknesses:**

I’m not an expert in theoretical analysis of convergence rate for the discrete diffusion model. Here’s how I understand it:

1. Quality and Clarity
The paper seems very solid and well-structured. The background is easy to understand, and the analysis is rigorous.  It would help if the authors explained more of the intuition behind their ideas, especially the surrogate initialisation method and what happens if you skip early stopping.

3. Significance:
The paper is important because absorbing discrete diffusion models are becoming more common in areas like language and molecule generation. The authors provide the first convergence guarantee to explain why these models work so well in practice, and this could guide future research and the practical use of these models.

4. Originality
The paper is novel since it focuses on absorbing rate matrices, which other papers haven’t analyzed in depth. The authors also come up with new ways to handle the unique challenges that come with these models, some of the techniques like  smooth surrogate distribution are intuitive and natural.

Limitation:
The paper lacks of empirical evidence which supports the derived theory.

---

> ### Author Rebuttal · Authors · 2025-07-30
>
> We thank the reviewer for carefully reading our work and providing encouraging and insightful feedback.
>
> **Q1:** Limitation: The paper lacks empirical evidence which supports the derived theory.
>
> **A1:** Please see our response in A3.
>
> **Q2:** There are some divergences which remain well-defined when the underlying distribution is a singleton or non a.c., e.g. the spread KL divergence. Do you think it is possible to use such divergence to get rid of the surrogate distribution?
>
> **A2:** We agree that spread-type divergences present a promising direction for potentially bypassing the surrogate initialization. It is worth noting, however, that applying spread-type divergences introduces its own challenges, as their rate-of-change w.r.t. time becomes hard to capture in the reverse CTMC process. Yet, we view this as an intriguing avenue for future work and thank the reviewer for highlighting it.
>
> **Q3:** While the theoretical contributions are strong, the paper currently lacks empirical verification or practical demonstrations showing how the proposed convergence bounds translate to real-world implementations. For instance, it would be valuable to see whether the improved convergence rates under absorbing rate matrices indeed yield faster or higher-quality sampling in practice. Some plots would be useful to support the theoretical argument.
>
> **A3:** We appreciate the reviewer’s interest in empirical validation. To clarify, this submission is focused on closing a theoretical gap by providing the first non-asymptotic convergence guarantees for discrete diffusion samplers with absorbing rate matrices, which previously lacked such analysis. That said, prior works — including [1] and [2] — have empirically demonstrated that absorbing rate matrices often yield better performance than uniform ones. Our theoretical findings thus offer formal justification for this empirically observed advantage.
>
> We thank the reviewer again for your comments. We hope that our responses resolved your concerns. Certainly, we are more than happy to answer your further questions if any.
>
> [1] Austin, Jacob et al. “Structured denoising diffusion models in discrete state-spaces.”
>
> [2] Lou, Aaron, et al. “Discrete diffusion modeling by estimating the ratios of the data distribution.”

---

> > ### Comment · Reviewer_gMRL · 2025-08-04
> > **Thanks**
> >
> > Thank the author for the detailed reply. It would be good to include the future direction in the revised paper. I do not have any concerns left.

---

> > > ### Author Response · Authors · 2025-08-04
> > >
> > > Definitely! We will include that in the revision. Thank you again for your comments and feedback.

---

### Decision · Program_Chairs · 2025-09-17

**Decision:**

Accept (poster)

**Comment:**

This work is a theoretical study of the discrete diffusion models with absorbing rate matrices. It establishes the first non-asymptotic convergence result in this setting. All reviewers agree the paper is well written and technically sound. There are two major concerns raised by the reviewers, one on assumption 4 and one on the significance of the results. The authors manage to convince the reviewers with an intense and comprehensive rebuttal. Overall, this work represents an interesting contribution in the area of discrete diffusion models.